# Generalized Interpolating Discrete Diffusion

**Dimitri von Rütte** [1]   **Janis Fluri** [1]   **Yuhui Ding** [1]   **Antonio Orvieto** [2,3]   **Bernhard Schölkopf** [1,2,3]   **Thomas Hofmann** [1]

## Abstract

While state-of-the-art language models achieve impressive results through next-token prediction, they have inherent limitations such as the inability to revise already generated tokens. This has prompted exploration of alternative approaches such as discrete diffusion. However, masked diffusion, which has emerged as a popular choice due to its simplicity and effectiveness, reintroduces this inability to revise words. To overcome this, we generalize masked diffusion, deriving a new family of general interpolating discrete diffusion (GIDD) which offers greater flexibility in the design of the noising processes. Leveraging a novel diffusion ELBO, we achieve compute-matched state-of-the-art performance in diffusion language modeling. Exploiting GIDD's flexibility, we explore a hybrid approach combining masking and uniform noise, leading to improved sample quality and unlocking the ability for the model to correct its own mistakes, an area where autoregressive models notoriously have struggled.
Code: https://github.com/dvruette/gidd/

## 1. Introduction

For certain data distributions such as natural images or natural language, the information content of any given sample can be overwhelming, making the task of generating realistic samples through generative modeling difficult. A common strategy to ease the burden on the generative model is to break up the task of generating an entire sample into multiple inference steps, each being simpler in isolation, but recovering the full distribution when recombined. The most prevalent example of this, especially for natural language, is autoregressive modeling (Bengio et al., 2000), where the task of generating a sentence (or sequence) is decomposed

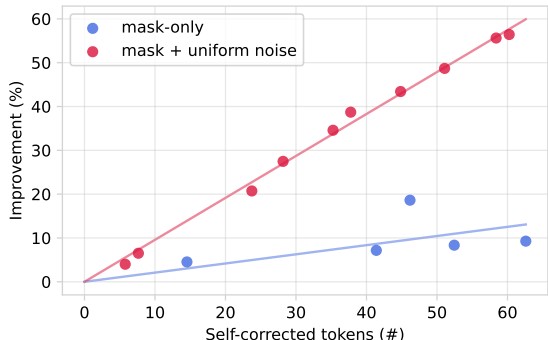

*Figure 1.* Training a diffusion model using GIDD on a combination of masking and uniform noise teaches it to identify and correct its own mistakes. By iteratively replacing bad tokens with better ones (as determined by the model), sample quality (as per generative PPL via Gemma 2 9B) improves by up to 55%.

into generating one word (or token) at a time, with each new word serving as additional context for the next word.

While extraordinarily successful on a wide range of data modalities (van den Oord et al., 2016a;b; Radford et al., 2018), there are some inherent challenges to this approach. First and most obviously, generating a sequence of length $N$ necessarily requires $N$ invocations of the model. This is not a problem if $N$ is small but can become expensive as $N$ grows to large numbers. Secondly, long-term dependencies and coherence can pose a challenge, for example if each step has a non-negligible error rate: If a wrong token is sampled or a previous token becomes incompatible with newly sampled tokens, there is no way to correct it. Considerable effort has gone into solving this limitation, most recently by post-training with reinforcement learning (RL) to teach sequential reasoning over multiple autoregressive steps (Bengio et al., 2015; Ranzato et al., 2015; Bahdanau et al., 2016; OpenAI et al., 2024; DeepSeek-AI et al., 2025).

Denoising diffusion models (Sohl-Dickstein et al., 2015) propose a different way of decomposing the generative task which can address both limitations. Instead of splitting the sample into elements of a sequence, we gradually decrease the information content of the entire sample by degrading it through the addition of some form of noise until, eventually, the information content reaches zero. The generative task then consists of reversing this degradation process, gradually adding information back in until the full sample is

---

[1]Data Analytics Lab, Department of Computer Science, ETH Zurich [2]ELLIS Institute Tübingen [3]Max Planck Institute for Intelligent Systems, Tübingen. Correspondence to: Dimitri von Rütte <dvruette@ethz.ch>.

*Proceedings of the $42^{nd}$ International Conference on Machine Learning*, Vancouver, Canada. PMLR 267, 2025. Copyright 2025 by the author(s).

| |
|---|
| Machine learning ~~are~~ is a field ~~to study~~ of research in artificial intelligence ~~during which~~ and the development [...] |
| ~~Republic of Delta~~World of Warcraft has made some significant improvements to ~~game~~ it in ~~their~~ the most recent ~~improvement~~ update, the ~~"Death in the Vengeance" change~~ "End of the World" update. |
| Mexico City is the largest city in ~~France~~ Mexico. With an estimated population of ~~22,752,000~~ 22,000,000 [...] |
| Suppose Alice has 5 apples. If Alice gives ~~2~~ all of ~~her~~ the apples to Bob, she is left with zero apples. |

*Table 1.* Examples of self-correction (green replaces red) by our GIDD+ BASE model trained with 20% uniform noise. The model is able to correct grammatical mistakes, improve word choices, and even improve factuality without being explicitly trained to do so.

restored. This decouples the number of model invocations from the size of the sample since the number of steps we take to fill in the missing information can be chosen freely. For natural images, an obvious and suitable degradation is the progressive addition of per-pixel Gaussian noise. This choice yields a simple training objective that works well in practice and forms the basis of state-of-the-art image generation models (Ho et al., 2020; Kingma et al., 2023). The success of image diffusion models has spurred interest in applications to other domains and modalities, including discrete data like text (Austin et al., 2023). Unfortunately, Gaussian diffusion cannot be naively applied to discrete data as there is not necessarily a notion of distance or similarity, at least not one that is straightforward to measure. Instead, discrete diffusion models have converged on a degradation process consisting of gradually removing ("masking") tokens until none are left (Austin et al., 2023; Shi et al., 2024; Sahoo et al., 2024). The task of the model then becomes to reconstruct the original sequence by "filling in the blanks" until all masked tokens have been filled in. This can also be thought of as autoregressive generation in a randomized order (Welleck et al., 2019), and indeed reintroduces one of its inherent limitations: Once a token is filled in, it can no longer be changed, and any intermediate errors will necessarily propagate to the final sample. However, by injecting a small amount of uniform token noise into the diffusion process, we can allow for any token to transition to any other token, therefore resolving this issue. This elicits the realization that the type of noise is an integral part of a diffusion model with potentially fundamental implications on its strengths and limitations. Drawing motivation from this, our work aims to illuminate the design space of discrete diffusion models by opening up the design space and exploring an alternative diffusion process that combines masking and uniform noise. Our contributions are two-fold:

On the theoretical side, in Section 3 we extend the framework of masked diffusion to general interpolating discrete diffusion (GIDD) processes. GIDD offers great flexibility in the choice of noising process, encompassing any diffusion process that can be written as a linear combination between the data and some (time-varying) mixing distribution. We derive closed-form solutions for the cumulative state transitions and the diffusion Evidence Lower Bound (ELBO)

for this general family, which are needed for sampling and likelihood training respectively. We also show that the derived ELBO has a global minimum that is reached when the model matches the true distribution.

On the practical side, in Sections 4 and 5 we apply our theory to the special case of masking noise in combination with varying levels of uniform noise. We conduct an ablation study, showing that our mask-only model achieves compute-matched state-of-the-art on diffusion language modeling thanks to a reweighted training objective (Sec. 5.2). We also show that the addition of uniform noise leads to improved sample quality and unlocks self-correction abilities (Fig. 1, Tab. 1) that allows the model to iteratively improve samples beyond what is possible by simply traversing the backward diffusion process (Sec. 5.4).

## 2. Discrete Diffusion Models

As the name suggests, discrete diffusion models act on a discrete state space $\mathcal{Z}$. Given some initial state $X \in \mathcal{Z}$ sampled from the data distribution $q_0(X)$, the sample is gradually degraded through a Markov chain $Z_1, \ldots, Z_T$ with $Z_t \in \mathcal{Z}$, $Z_{t+1} \sim q_t(Z_{t+1}|Z_t)$, and $Z_1 = X$ until reaching some (easy-to-sample) prior distribution $p_T(Z_T)$. The denoising task then becomes to learn the backward kernel of this Markov chain, such that we can (approximately) reverse the degradation process for any $Z_T$ sampled from the prior distribution. Oftentimes, the state space is structured as a sequence (of length $L$) of tokens from a vocabulary $V$, i.e. $Z_t = (z_t^{(1)}, \ldots, z_t^{(L)})$ with $z_t^{(i)} \in V$. In this case, it is common to add noise to each token independently such that it suffices to look at the forward and backward noising trajectory of any token $z_t$ in isolation. This is possible if the initial state $X$ is known, which it is during training but not during inference. The model must therefore learn to make predictions without this knowledge, inferring as much as possible about $X$ from its noisy version, the sequence $Z_t$.

### 2.1. Interpolating Masked Diffusion

Masked diffusion models (MDM) have seen widespread adoption by the community (Ou et al., 2024; Shi et al., 2024; Sahoo et al., 2024; Nie et al., 2024; Hu & Ommer, 2024) due to their simplicity and good performance. The

core idea is to progressively replace tokens with a special [MASK] token until every token has been replaced. As such, the denoising task for the model to learn is to "fill in the blanks" given some context. This noising process results in a Markov chain with marginal transitions that can be written as a linear interpolation between mask and data:

$$q_t(z_t|x) = \mathrm{Cat}(z_t; \alpha_t \mathbf{x} + \beta_t \mathbf{m}), \tag{1}$$

where $\beta_t = 1 - \alpha_t$, $\mathbf{x}$ and $\mathbf{m}$ denote the one-hot encoding of the data $x$ and the masking token $m$ respectively,[1] and $0 \leq \alpha_t \leq 1$ determines the signal-to-noise ratio (SNR) at the current time $t$. The Evidence Lower Bound (ELBO) of MDM takes the form of a simple weighted reconstruction loss of the missing tokens. Specifically, with $\mathbf{x}_\theta$ denoting a neural network that predicts the distribution of $x$ given a partially noised sequence $Z_t = (z_t^{(1)}, \ldots, z_t^{(L)})$, the negative ELBO is given by

$$
\begin{aligned}
&- \log p(x) \\
&\leq \mathbb{E}_{t,z_t} \left[ \frac{\alpha'_t}{1 - \alpha_t} \delta_{z_t,m} \mathbf{x}^\top \log \mathbf{x}_\theta(Z_t, t) \right] + C, \quad (2)
\end{aligned}
$$

where $t \sim \mathcal{U}(0,1)$ and $z_t \sim q_t(z_t|x)$, with $\delta$ denoting the Kronecker delta function. Recall that the input to $\mathbf{x}_\theta$ is the entire noisy sequence $Z_t$ whereas everything else happens for each token independently.

## 2.2. Limitations of Masked Diffusion

Despite their popularity, MDMs have some fundamental limitations. Most obviously: due to the way the underlying Markov chain is defined, a token can never be changed again once it has been filled in, which is analogous to autoregressive prediction. This can lead to the accumulation of errors or some tokens becoming incompatible as more tokens are unmasked, and with no way to fix them, they inadvertently persist to the final result. Another, less severe limitation is the fact that only masked tokens carry a loss signal, as unmasked tokens are always completely noise-free. Like with BERT, this results in a smaller effective batch size which can lead to slower convergence compared to autoregressive models (Devlin et al., 2019; Clark et al., 2020).

## 3. Generalized Interpolating Diffusion

To resolve these limitations, we would like to expand our horizon to a more diverse set of diffusion processes. A natural solution drawing inspiration from BERT (Devlin et al., 2019) would be to use a combination of masking and uniform noise. This would address both limitations described above: Not only do we gain the ability to change already-unmasked tokens during sampling, but we also obtain a more informative training task, as every token in the

sequence (whether masked or not) could potentially be corrupted and thus require correction. With the model learning to distinguish between "correct" and "incorrect" tokens, it may also learn to correct its own mistakes, a notion that will be confirmed in Section 5.4.

However, there are some technical challenges to training a diffusion model on some specific, desirable diffusion trajectory. The canonical training objective, the diffusion ELBO, cannot be derived without knowledge of the Markovian state transitions, but crafting a Markov chain with specific emerging properties (e.g. "halfway in the diffusion process, 40% of tokens should be masked, 40% should be unperturbed, and 20% should be random") is generally a non-trivial inverse problem. Instead of solving this inverse problem for a specific combination of masking and uniform noise, and to gain the necessary flexibility to design an effective model, we aim to generalize interpolating diffusion from mask-only to arbitrary (time-varying) interpolants. Specifically, we introduce the Generalized Interpolating Discrete Diffusion process (GIDD), a family of diffusion models with marginal forward transitions

$$q_t(z_t|x) = \mathrm{Cat}(z_t; \alpha_t \mathbf{x} + \beta_t \boldsymbol{\pi}_t), \tag{3}$$

where $\boldsymbol{\pi}_t$ can be any probability distribution that changes smoothly over time. Notably, masked diffusion is a special case of GIDD for $\boldsymbol{\pi}_t = \mathbf{m}$. We will show the existence of a Markov chain that results in these marginals for any suitable $\alpha_t$ and $\boldsymbol{\pi}_t$ and derive its conditional transitions as well as the associated ELBO necessary for likelihood training.

### 3.1. Forward Process

GIDD is designed to allow maximal flexibility over the type of noise added to the data at any point in time. It consists of a mixing rate $\alpha_t$, which defines the signal-to-noise ratio over time, and a mixing distribution $\boldsymbol{\pi}_t$, which defines what distribution the data is noised towards at any given time. We refer to the combination of these two functions as the "mixing schedule" of our diffusion process.

**Definition 3.1** (Mixing Rate). Let the (cumulative) mixing rate $\alpha_t, \beta_t$ with $\beta_t = 1 - \alpha_t$ be a time-differentiable decreasing function $\alpha_t : [0,1] \mapsto [0,1]$ where the initial value $\alpha_0 = 1$ means no mixing and the final value $\alpha_1 = 0$ is complete mixing. This determines the SNR with $\mathrm{SNR} = \alpha_t/\beta_t$.

**Definition 3.2** (Mixing Distribution). Let the mixing distribution $\boldsymbol{\pi}_t$ be a time-dependent probability vector, i.e. a time-differentiable function $\boldsymbol{\pi}_t : [0,1] \mapsto \Delta^{|V|-1}$ where $\Delta^{|V|-1}$ denotes the $|V|$-dimensional simplex.[2] The distribution $\boldsymbol{\pi}_t$ determines the type of noise that is added to the data at any time $t$. As a consequence, $\boldsymbol{\pi}_1$ represents the prior distribution of our diffusion process.

---

[1] Throughout, we will use bold letters to denote vectors, which, in reference to a token, denotes the one-hot encoding of the token.

[2] The $d$-dimensional simplex $\Delta^{d-1}$ is defined as the set of all points $x \in \mathbb{R}^d$ with $x_i \geq 0$ and $\sum_i^d x_i = 1$.

Ultimately, we want to find a diffusion Markov chain with marginals as postulated in Equation (3), but to arrive at this conclusion we will have to work our way up from the underlying discrete-time Markov chain to the continuous-time state transitions, to the closed-form cumulative transitions.

**Proposition 3.3** (GIDD Conditional Transitions). *Let $\alpha_t$, $\beta_t = 1 - \alpha_t$ denote the mixing rate and let $\boldsymbol{\pi}_t$ denote the mixing distribution. Then there exists a continuous-time Markov chain with transition probabilities from state $z_s$ to $z_t$ at times $s \leq t$ given by*

$$q_{t|s}(z_t|z_s) = \mathrm{Cat}(z_t; Q_{t|s}\mathbf{z}_s), \quad Q_{t|s} = \alpha_{t|s}I + \beta_{t|s}\boldsymbol{\pi}_{t|s}\mathbf{1}^\top, \tag{4}$$

*where $\alpha_{t|s} = \frac{\alpha_t}{\alpha_s}$, $\beta_{t|s}\boldsymbol{\pi}_{t|s} = \beta_t\boldsymbol{\pi}_t - \frac{\alpha_t}{\alpha_s}\beta_s\boldsymbol{\pi}_s$, and $\mathbf{1}$ denotes the $|V|$-dim. vector of all ones.*

*Proof.* Let us discretize time into a $\Delta$-spaced mesh for some arbitrary $\Delta > 0$, i.e. assume that we can write $t = \Delta i$ with $i \in \mathbb{Z}$ for any $t$. We then define the instantaneous mixing schedule[3] $\dot{\alpha}_t$ and $\dot{\beta}_t\dot{\boldsymbol{\pi}}_t$ as

$$\dot{\alpha}_{\Delta i} = \frac{\alpha_{\Delta(i+1)}}{\alpha_{\Delta i}}, \tag{5}$$

$$\dot{\beta}_{\Delta i}\dot{\boldsymbol{\pi}}_{\Delta i} = \beta_{\Delta(i+1)}\boldsymbol{\pi}_{\Delta(i+1)} - \frac{\alpha_{\Delta(i+1)}}{\alpha_{\Delta i}}\beta_{\Delta i}\boldsymbol{\pi}_{\Delta i}. \tag{6}$$

The instantaneous transition probability is now defined as

$$\dot{q}_t(z_{t+\Delta}|z_t) = \mathrm{Cat}(z_{t+\Delta}; \dot{Q}_t\mathbf{z}_t), \quad \dot{Q}_t = \dot{\alpha}_tI + \dot{\beta}_t\dot{\boldsymbol{\pi}}_t\mathbf{1}^\top. \tag{7}$$

The instantaneous transitions induce a discrete-time Markov chain with the desired mixing properties as defined by our mixing schedule.

We now turn to our main objective: the cumulative transition matrix $Q_{t|s}$ of this Markov chain, which is defined as $Q_{t|s} = \prod_{i=s/\Delta}^{t/\Delta-1} \dot{Q}_{\Delta i}$. We need to show that $Q_{t|s} = \alpha_{t|s}I + \beta_{t|s}\boldsymbol{\pi}_{t|s}\mathbf{1}^\top$. To this end, we are going to inductively unroll a single step to find recursive formulas for $\alpha_{t|s}$ and $\beta_{t|s}\boldsymbol{\pi}_{t|s}$. First, note that the base case $t = s$ is simply $Q_{s|s} = I$ with $\alpha_{s|s} = 1$ and $\beta_{s|s}\boldsymbol{\pi}_{s|s} = 0$ as we must remain in the same state. Next, assume that the induction hypothesis holds for $Q_{t|s}$. We then have

$$Q_{t+\Delta|s} = \dot{Q}_tQ_{t|s} \tag{8a}$$

$$= [\dot{\alpha}_tI + \dot{\beta}_t\dot{\boldsymbol{\pi}}_t\mathbf{1}^\top] \cdot [\alpha_{t|s}I + \beta_{t|s}\boldsymbol{\pi}_{t|s}\mathbf{1}^\top] \tag{8b}$$

$$= \dot{\alpha}_t\alpha_{t|s}I + \dot{\beta}_t(\alpha_{t|s}\dot{\boldsymbol{\pi}}_t\mathbf{1}^\top I + \beta_{t|s}\dot{\boldsymbol{\pi}}_t(\mathbf{1}^\top\boldsymbol{\pi}_{t|s})\mathbf{1}^\top) + \dot{\alpha}_t\beta_{t|s}\boldsymbol{\pi}_{t|s}\mathbf{1}^\top \tag{8c}$$

$$= \dot{\alpha}_t\alpha_{t|s}I + \dot{\beta}_t(\alpha_{t|s} + \beta_{t|s})\dot{\boldsymbol{\pi}}_t\mathbf{1}^\top + \dot{\alpha}_t\beta_{t|s}\boldsymbol{\pi}_{t|s}\mathbf{1}^\top \tag{8d}$$

$$= \underbrace{\dot{\alpha}_t\alpha_{t|s}}_{=\alpha_{t+\Delta|s}}I + (\underbrace{\dot{\beta}_t\dot{\boldsymbol{\pi}}_t + \dot{\alpha}_t\beta_{t|s}\boldsymbol{\pi}_{t|s}}_{=\beta_{t+\Delta|s}\boldsymbol{\pi}_{t+\Delta|s}})\mathbf{1}^\top, \tag{8e}$$

[3]Note that while we use the dot-notation ($\dot{\alpha}$) for instantaneous changes, this is not to be confused with the time-derivative, which we denote by a prime ($\alpha'$).

where we use the fact that $\mathbf{1}^\top\boldsymbol{\pi}_{t|s} = 1$ and $\alpha_{t|s} + \beta_{t|s} = 1$ (as per Lemma H.1, App. H.1). Having found recursive formulas for $\alpha_{t|s}$ and $\beta_{t|s}\boldsymbol{\pi}_{t|s}$ we can now apply telescoping to find the desired closed-form solutions (see App. H.2 for details), proving the original claim for any $\Delta > 0$. In particular, the proof also holds in the limit of $\Delta \to 0$ as long as the limits $\lim_{\Delta \to 0}\dot{\alpha}_{\Delta i}$ and $\lim_{\Delta \to 0}\dot{\beta}_{\Delta i}\dot{\boldsymbol{\pi}}_{\Delta i}$ exist. Differentiability of $\alpha_t$ and $\boldsymbol{\pi}_t$, as required by Definitions 3.1 and 3.2, is sufficient for this. $\qquad\square$

**Corollary 3.4.** *The cumulative transition probabilities of the Markov chain from Proposition 3.3 are given by*

$$q_t(z_t|x) = \mathrm{Cat}(z_t; Q_t\mathbf{x}), \quad Q_t = \alpha_tI + \beta_t\boldsymbol{\pi}_t\mathbf{1}^\top. \tag{9}$$

*Proof.* The claim follows directly from Proposition 3.3 with $Q_t = Q_{t|0}$ and using the fact that $\alpha_0 = 1$, $\beta_0 = 0$. $\qquad\square$

With this, we have successfully constructed a Markov chain with the desired marginals outlined in Equation 3. For deriving the ELBO later on, we also need the transition rates of the corresponding Continuous-Time Markov Chain (CTMC), which is defined as follows.

**Definition 3.5** (CTMC Forward Transition). For some start time $s$ and end time $t = s + \Delta$ with $\Delta \to 0$, we have

$$q_{t|s}(z_t|z_s) = \delta_{z_s,z_t} + R_t(z_s, z_t)\Delta + o(\Delta), \tag{10}$$

where $R_t$ is called the forward transition rate. Little-$o$ notation is used for denoting asymptotically sub-linear terms.

We now characterize the CTMC forward rate of GIDD.

**Lemma 3.6** (GIDD Forward Rate). *The CTMC forward rate matrix $R_t$ of GIDD is given by*

$$R_t(z_s, z_t) = \frac{\alpha_t'}{\alpha_t}\delta_{z_s,z_t} + \mathbf{z}_t^\top\left(\beta_t\boldsymbol{\pi}_t' - \frac{\alpha_t'}{\alpha_t}\boldsymbol{\pi}_t\right), \tag{11}$$

*where $\alpha_t'$ and $\boldsymbol{\pi}_t'$ denote the time-derivative of the respective mixing function.*

*Proof.* By performing a first-order Taylor expansion on $q_{t|s}(z_t|z_s)$ and rearranging the result, we arrive at the desired expression. See Appendix H.3 for details. $\qquad\square$

### 3.2. Backward Process

We choose the same parameterization of the backward process as prior work (Sohl-Dickstein et al., 2015; Austin et al., 2023). This canonical form of the model distribution $p_\theta(z_s|z_t)$ is given by

$$p_\theta(z_s|z_t) = q_{t|s}(z_t|z_s)\frac{q_s(z_s|\mathbf{x}_\theta)}{q_t(z_t|\mathbf{x}_\theta)}, \tag{12}$$

with shorthand notation $q_t(z_t|\mathbf{x}_\theta) := \mathrm{Cat}(z_t; Q_t\mathbf{x}_\theta(Z_t, t))$, where $\mathbf{x}_\theta(Z_t, t)$ is a neural network that predicts the distribution of $x$ given the noised sequence $Z_t$. We refer to Appendix H.4 for details on the CTMC backward rate $\hat{R}_t(z_t, z_s)$, which is also required for the ELBO derivation.

### 3.3. ELBO

In order to train a GIDD model, we need a differentiable way to estimate its likelihood. The Evidence Lower Bound (ELBO) serves this purpose: By maximizing a lower bound, we implicitly also maximize the (worst-case) likelihood of our model. For the ELBO, we need the forward and backward rate of GIDD, which we have already derived. Then, starting with a slightly modified version of the ELBO from Campbell et al. (2022), we plug in our forward and backward rates $R_t(z_s, z_t)$ and $\hat{R}_t(z_t, z_s)$ and simplify to obtain Theorem 3.7 (complete proof in App. H.5).

**Theorem 3.7** (GIDD ELBO). *Let $\alpha_t$, $\beta_t$ and $\boldsymbol{\pi}_t$ be a mixing schedule as defined in Definitions 3.1, and 3.2 with marginal forward distribution $q_t(z_t|x)$ as defined in Equation 3. Let further $w_t(z_t, x)$ be a weighting function defined as*

$$w_t(z_t, x) = \frac{1}{q_t(z_t|x)} \mathbf{z}_t^\top \left( \beta_t \boldsymbol{\pi}'_t - \frac{\alpha'_t}{\alpha_t} \boldsymbol{\pi}_t \right). \quad (13)$$

*Then, the continuous-time negative ELBO (CT-NELBO) of the corresponding diffusion model is given by*

$$
\begin{aligned}
-\log p(x) \leq \mathbb{E}_{t,z_t}[w_t(z_t, x)(D_{KL}(q_t(\cdot|x)\|q_t(\cdot|\mathbf{x}_\theta)) \\
+ D_{IS}(q_t(z_t|x)\|q_t(z_t|\mathbf{x}_\theta)))] + C, \quad (14)
\end{aligned}
$$

*where $D_{IS}$ is the (pointwise) Itakura-Saito divergence defined as $D_{IS}(p\|q) = p/q - \log p/q - 1$, $t \sim \mathcal{U}(0,1)$, and $z_t \sim q_t(\cdot|x)$, and with $C$ denoting the ELBO constant*

$$C = \mathbb{E}_{q_0(z_0|x)}[\log p(x|z_0)] - D_{KL}(q_1(z_1|x)\|p_1(x_1)). \quad (15)$$

Since GIDD is a strictly more general form of the widely used masked diffusion paradigm (Ou et al., 2024; Shi et al., 2024; Sahoo et al., 2024), we expect the canonical MDM ELBO to emerge from the GIDD ELBO by choosing an appropriate mixing schedule, which is indeed what we find by setting $\boldsymbol{\pi}_t = \mathbf{m}$ (proof in App. H.9).

**Corollary 3.8** (Equivalence to MDM). *If $\boldsymbol{\pi}_t = \mathbf{m}$, then, for any valid noise schedule $\alpha_t$, the GIDD ELBO reduces to the MDM ELBO (Eq. 2).*

**Interpretation.** Taking a closer look at the GIDD ELBO, we notice that it consists of solving two tasks jointly:

1) The first task is to match the model to the marginal forward distribution of some sample $x$ given its noised version $z_t$ by minimizing the KL-divergence between the two distributions at the current noise level.

2) The second task is to minimize the pointwise IS-divergence at between the model and the true marginal distribution *at the sampled $z_t$*.

Since both tasks consist of minimizing a divergence between

the model and the true distribution, they are both minimal if and only if $q_t(\cdot|x) = q_t(\cdot|\mathbf{x}_\theta)$, implying that the ELBO is minimal there.[4] Indeed, it can be shown that the global minimum of the ELBO is reached only at that point.

**Proposition 3.9.** *For any mixing schedule $\alpha_t$ and $\boldsymbol{\pi}_t$, the GIDD CT-NELBO has a global minimum of zero (up to the ELBO constant $C$), which is reached if and only if $q_t(z_t|x)$ and $q_t(z_t|\mathbf{x}_\theta)$ are the same everywhere.*

*Proof.* See Appendix H.6. □

This is good news, since it tells us that the mixing schedule theoretically does not limit the best-possible model.

In conclusion, the GIDD ELBO is a straight-forward and flexible training objective that can be applied out-of-the-box to any interpolating diffusion model.

### 3.4. Sampling

Given some sampling schedule $0 \approx t_0 < t_1 < \cdots < t_T \approx 1$ and some neural network $\mathbf{x}_\theta$, we employ ancestral sampling by discretizing time along the chosen mesh. Specifically, starting with a sequence of all mask tokens, i.e. $z_{t_T} = m$ for all $z_{t_T}$, we iteratively sample $p_\theta(z_{t_{i-1}}|z_{t_i})$ for $i = T, \ldots, 1$:

$$z_{t_{i-1}} \sim q_{t_i|t_{i-1}}(z_{t_i}|z_{t_{i-1}}) \frac{q_{t_{i-1}}(z_{t_{i-1}}|\mathbf{x}_\theta(Z_{t_i}, t_i))}{q_{t_i}(z_{t_i}|\mathbf{x}_\theta(Z_{t_i}, t_i))} \quad (16)$$

**Self-Correction Step.** In addition, we propose a fixed-point iteration to improve generated samples by resampling some tokens according to the model's judgement. More precisely, we give the fully denoised sample $Z_{t_0}$ to the model and sample the resulting distribution with some temperature $\tau$. Then, of all sampled tokens different from $Z_{t_0}$, we select the one with the highest model likelihood and commit it. This is repeated until convergence (details in App. C).

## 4. Mixing Schedule

While GIDD can be used for masked diffusion, our original motivation for introducing a generalized framework was to explore the combination of masking and uniform noise. To this end, we design a mixing schedule that keeps the masked prior distribution but allows for configurable amounts of uniform noise in between. We use $p_u$ to denote the amount of uniform noise: For the sake of interpretability, the expected fraction of uniform tokens should reach a maximum of $p_u$ at the midpoint between data and noise ($t = 1/2$). With these desiderata in mind, we define our mixing rate and mixing distribution (Def. 3.1 and 3.2):

$$\alpha_t = \frac{1-t}{C_t}, \quad \beta_t \boldsymbol{\pi}_t = \frac{t}{C_t}\mathbf{m} + \frac{c_t}{C_t}\mathbf{u}, \quad (17)$$

---

[4]It is worth noting that both the KL and the IS-divergence are Bregman divergences, implying that they can be linearly combined into a single Bregman divergence $D_F$ with $F(p|z_t) = \sum_z p_z \log p_z - \log p_{z_t}$.

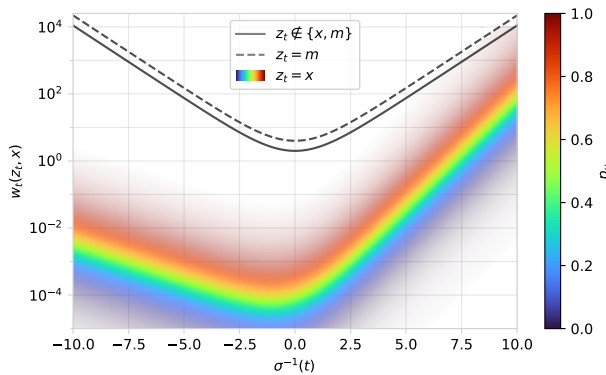

*Figure 2.* ELBO weights grow exponentially for very low/high noise levels, causing poor optimization if not handled carefully. While masked and uniform token weights are almost constant, noise-free token weights vary heavily depending on $p_u$.

where $\mathbf{u} = \frac{1}{N-1}(\mathbf{1} - \mathbf{m})$ denotes the uniform probability vector, $c_t = Bt^{\frac{\gamma}{2}}(1 - t)^{\frac{\gamma}{2}}$, $C_t = 1 + c_t$, $N$ is the vocabulary size, and $B$ is a constant chosen such that the desired uniform token ratio is reached. The marginal forward distribution then becomes

$$q_t(z_t|x) = \frac{1}{C_t}((1-t)\mathbf{x} + t\mathbf{m} + c_t\mathbf{u}). \qquad (18)$$

To reach the desired uniform noise level $p_u$ at $t = 1/2$, we need to set $B = 2^\gamma \frac{p_u}{1-p_u}$ (proof in App. H.7). The GIDD ELBO weights $w_t(z_t, x)$ can also be derived in closed form (see App. H.8 for details). Note that setting $p_u = 0.0$ again recovers masked diffusion. Finally, we set $\gamma = 1$, but there may be other valid choices for this and other variables introduced in this section.

### 4.1. Training Objective

Before starting our experiments, we need to solve one last issue, which will yield great performance gains. Taking a closer look at the ELBO weights $w_t(z_t, x)$, we find that their behavior for $t \to 0$ and $t \to 1$ is quite extreme. Consider the three possible cases $z_t = x$, $z_t = m$, and $z_t \notin \{x, m\}$. Plotting the weights $w_t(z_t, x)$ over time[5] reveals that the weight grows exponentially for very low/high noise levels in all three cases (Figure 2). This can be problematic since these low/high noise samples provide little to no training signal as the model's job of denoising becomes either trivial or impossible, yet can easily drown out all other samples in the batch. To counteract this issue, we propose two weighting schemes that reduce the influence of extreme samples, hence emphasizing intermediate noise levels where the training task is informative.

The simple and obvious solution is to clamp the weights to

---

[5] $\sigma^{-1}(t)$ denotes the inverse sigmoid function and can be thought of as the (negative) log-SNR when ignoring the effect of $C_t$, which tends to be negligible for small $p_u$.

| Model (SMALL) | Train. toks. | PPL ($\downarrow$) |
|---|---|---|
| *Autoregressive* | | |
| GPT2 (Radford et al., 2019) | unk. | 23.40 |
| Llama 110M (retrain.) | 262B | 16.11 |
| *Diffusion* | | |
| MD4[*] (Shi et al., 2024) | 524B | 21.80 |
| MDLM[*] (Sahoo et al., 2024) | 262B | 23.21 |
| MDM (reimpl.) | 262B | 23.36 |
| GIDD+ (ours; $p_u = 0.0$) | 262B | 22.29 |

*Table 2.* Our best GIDD model outperforms the compute-matched MDM (reimpl.) baseline, which in turn closely matches results from the MDM literature in terms of validation PPL on OWT. [*]Numbers reported by the original paper.

| Model (SMALL) | PPL ($\downarrow$) | | |
|---|---|---|---|
| | $p_u = 0.0$ | $p_u = 0.1$ | $p_u = 0.2$ |
| MDM (reimpl.) | 24.37 | - | - |
| GIDD (ours) | 24.36 | 26.88 | 28.22 |
| + weight clipping | 23.23 | 25.09 | 26.40 |
| + dynamic weights | 23.24 | 23.90 | 24.64 |
| + weight decay | **23.05** | **23.67** | **24.38** |

*Table 3.* PPL of GIDD ($p_u = 0.0$) and MDM match closely, as expected from their theoretical equivalence. Significant gains come from choosing the right weighting function, especially in the $p_u > 0$ regime. The final best setting includes dynamic loss weights $\tilde{w}_t^{\mathrm{dyn}}$ and weight decay and is also referred to as *GIDD+*.

some maximal value $w_{\max}$, so we define

$$\widetilde{w}_t^{\mathrm{clamp}}(z_t, x) = \min(w_{\max}, w_t(z_t, x)). \qquad (19)$$

Through preliminary experiments, we find $w_{\max} = 1$ to be best, so this is used throughout. Note that clamping mostly affects the weights of mask and uniform tokens. A more principled approach may aim to keep the maximum loss weight constant while preserving the relative weights between masked, uniform, and noise-free tokens. We call this the dynamic weighting function and define it as

$$\widetilde{w}_t^{\mathrm{dyn}}(z_t, x) = w_{\max}(1 + \delta_{z_t, m} + (\tfrac{B}{N}e^{-\frac{\lambda_t}{2}} - 1)\delta_{z_t, x}), \quad (20)$$

where $\lambda_t = \log \frac{\alpha_t}{1-\alpha_t}$ is the log-SNR. The relative weights ($2 \,/\, 1 \,/\, \frac{B}{N}e^{-\frac{\lambda_t}{2}}$) are determined empirically. Note that reweighting the ELBO like this is equivalent to sampling $t$ from a non-uniform distribution or choosing a different noise schedule during training (Kingma & Gao, 2023).

## 5. Experiments

### 5.1. Experimental Setup

While discrete diffusion models are a natural fit for any discrete data, we focus our attention specifically on language

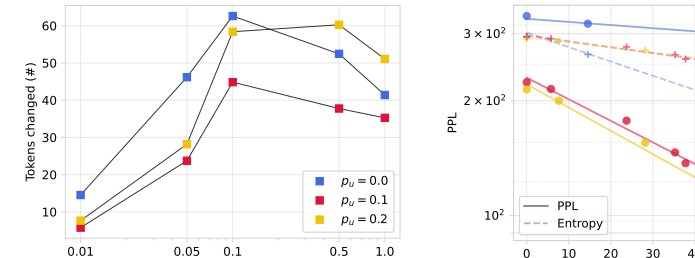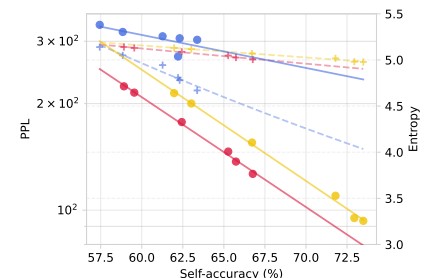

*Figure 3.* From left to right: (a) Self-correction using GIDD+ (BASE) models resamples up to 10% of tokens independent of the uniform noise level. A temperature of $\tau \in [0.1, 0.5]$ is found to be most effective. (b) For models trained on hybrid noise, sample quality (PPL) improves significantly as more tokens are changed. The mask-only model, though, is unable to improve quality despite resampling as many tokens. Sample diversity (entropy) drops noticeably for mask-only models, but only slightly for hybrid models. (c) The correlation between self-accuracy and generative PPL reveals that hybrid models are significantly better at judging the quality of their own samples.

| Model | Clarity | Grammaticality | Factuality | Writing style | Creativity |
|---|---|---|---|---|---|
| GIDD ($p_u = 0.0$) | 2.51 | 2.96 | 3.61 | 2.84 | **4.48** |
| + self-corr. ($\tau = 0.1$) | 1.99 (-20.9%)** | 2.39 (-19.3%)** | 3.02 (-16.2%)** | 2.24 (-21.1%)** | 3.60 (-19.5%)** |
| GIDD ($p_u = 0.1$) | 2.51 | 2.85 | 3.66 | 2.78 | 4.26 |
| + self-corr. ($\tau = 0.1$) | 2.69 (+7.2%)** | 3.05 (+6.9%)** | 3.88 (+6.0%)** | 2.98 (+7.1%)** | 4.35 (+2.1%)* |
| GIDD ($p_u = 0.2$) | 2.49 | 2.82 | 3.70 | 2.79 | 4.25 |
| + self-corr. ($\tau = 0.5$) | **2.90** (+16.5%)** | **3.29** (+16.6%)** | **4.01** (+8.5%)** | **3.16** (+13.4%)** | **4.48** (+5.5%)** |

*Table 4.* Self-correction significantly improves various quality aspects as judged by GPT-4o on a scale from 1–10, but only for models trained with hybrid uniform noise. Applying self-correction in the mask-only setting is detrimental across the board. The highest level of uniform noise has the biggest improvements and highest scores across all categories. *$> 2\sigma$ difference, **$> 5\sigma$ difference.

modeling as it is one of the most prevalent tasks in modern machine learning. To this end, we adopt the OpenWebText (OWT) dataset (Gokaslan et al., 2019) since there exists a rich literature for both autoregressive and diffusion models trained on this dataset. We follow prior work (Sahoo et al., 2024; Shi et al., 2024) in terms of architecture and training scale and use the DiT architecture (Peebles & Xie, 2023) with the GPT2 tokenizer (Radford et al., 2019) and train SMALL (110M) and BASE (320M) models on 131B or 262B tokens, depending on the experiment (details in App. E).

### 5.2. Ablation Study

The goal of our ablation study is to answer three main questions: 1) Does GIDD with our mixing schedule and $p_u = 0.0$ recover MDM as theory predicts? 2) How does the addition of uniform noise affect performance? And 3) what is the importance of the weighting function (Sec. 4.1)?

To this end, we train SMALL GIDD models on OWT with varying levels of uniform noise $p_u \in \{0.0, 0.1, 0.2\}$. We also train our reimplementation of MDM on the same setup. The final validation perplexity (PPL) of these runs is reported in Table 3. We find that the training trajectories as well as the final performance of MDM and GIDD ($p_u = 0.0$) match almost perfectly with a respective validation PPL of 24.37 and 24.36. Our MDM reimplementation also closely matches the compute-matched MDLM (Sahoo et al., 2024)

baseline (Tab. 2) considering the slight differences in hyper-parameters.

However, adding uniform noise to the diffusion process, we find that perplexity degrades slightly, yet benefiting expressivity as we will see later (Sec. 5.3 and 5.4). This difference likely stems from an increase in task complexity: The combination of masking and uniform noise requires solving multiple tasks jointly, which is strictly more difficult and likely requires more capacity. This is supported by the observation that all noise levels scale consistently with model size, with the highest noise setting even showing some signs of improved scaling behavior (App. A).

Our custom weighting schemes bring non-trivial performance gains to both the mask-only and hybrid noise settings, and particularly the dynamic weighting scheme $\widetilde{w}_t^{\mathrm{dyn}}$ closes the gap significantly. We hypothesize that the difference between $\widetilde{w}_t^{\mathrm{clamp}}$ and $\widetilde{w}_t^{\mathrm{dyn}}$ is due to the importance of noise-free tokens, which have zero weight if $p_u = 0.0$ but cannot be ignored otherwise. Therefore, keeping the true relative weights between different token types seems to be beneficial if $p_u > 0$. Finally, a moderate amount of weight decay (0.02) improves both training and validation loss, as suggested by D'Angelo et al. (2024). The best configuration uses dynamic loss weights $\widetilde{w}_t^{\mathrm{dyn}}$ and 0.02 weight decay, which we refer to as GIDD+ from hereon out.

| Size | Model | Train. toks. | ARC-e | ARC-c | BoolQ | Hellaswag | PIQA | OBQA | WinoG. | Avg. |
|------|-------|--------------|-------|-------|-------|-----------|------|------|--------|------|
| SMALL | GPT2 | unk. | **43.81** | 19.03 | 48.72 | 28.92 | **62.89** | 16.40 | 51.62 | 38.77 |
| | Llama (retrain.) | 262B | 40.53 | **25.51** | 46.21 | **33.14** | 62.73 | **28.40** | 50.75 | **41.04** |
| | MDM (reimpl.) | 262B | 30.98 | 23.63 | 50.52 | 31.11 | 54.13 | 28.00 | 49.41 | 38.25 |
| | GIDD+ ($p_u = 0.0$) | 262B | 30.98 | 23.55 | 50.43 | 31.87 | 56.42 | 26.60 | 51.70 | 38.79 |
| | GIDD+ ($p_u = 0.0$) | 131B | 31.57 | 24.57 | **50.92** | 31.36 | 56.31 | 27.80 | **52.57** | 39.30 |
| BASE | GIDD+ ($p_u = 0.0$) | 131B | 32.58 | 24.40 | 50.86 | 36.62 | 58.05 | 29.2 | 51.54 | 40.46 |

*Table 5.* Our best GIDD+ model in terms of zero-shot benchmark accuracy outperforms MDM (reimpl.) and even surpasses GPT2-small, although it still lags behind our Llama-based autoregressive baseline. Best scores among SMALL models and diffusion models are bolded and underlined respectively.

## 5.3. Unconditional Generation

While models trained with uniform noise consistently exhibit a higher loss, we have yet to test the main motivation for its addition: By teaching the model to distinguish between "correct" and "incorrect" tokens, we hope to unlock the ability for the model to correct its own mistakes at generation time, stabilizing the denoising process and yielding improved sample quality. In order to quantify sample quality, we use "generative perplexity", a metric that computes the likelihood of generated samples under a more capable model (in our case Gemma 2 9B, Gemma Team (2024)), where a high likelihood under the reference model is considered to be a sign of high quality. While this metric has its flaws (see App. G), it is common in the literature (Lou et al., 2024; Sahoo et al., 2024). We find that, though absolute numbers are difficult to interpret in isolation, it is still useful for comparing models in relative terms, especially if controlling for diversity. To that end, we also consider the unigram entropy of generated samples as a diversity signal, which should stay close to the entropy of the data (4.98).

Notably, the generative PPL of models trained on uniform noise is significantly better than that of mask-only models, with entropy hovering around 5.15 for all models and settings (App. D). We observe especially big improvements over mask-only models for low inference-compute settings, with a generative PPL of 387 for GIDD+ (SMALL; $p_u = 0.1$) at 32 denoising steps compared to 904 for $p_u = 0.0$ and 1302 for MDM (App. D). Training on uniform noise therefore seems to stabilize the generation process when the model gets its own outputs as subsequent inputs, resulting in better sample quality despite having a slightly worse validation PPL. This suggests that some amount of self-correction may already be happening during the denoising process. However, while more denoising steps monotonically improve sample quality, this plateaus at a PPL of around 200 for BASE models (App. D). Next, we show that it is possible to decrease generative PPL well below this plateau by further exploiting the models' capabilities.

## 5.4. Self-Correction

In order to directly evaluate the model's self-correction abilities, we iteratively apply the self-correction step from Section 3.4 to unconditionally generated samples. If the model indeed has learned to identify and correct mistakes, including its own, we expect that this repeated invocation can iteratively improve samples until a stable point is reached where the model is either happy with the sample or sees no way to improve it. To measure the degree of convergence, or how "happy" the model is with a sample, we use its self-accuracy on the given sample, i.e. the percentage of tokens that have maximal likelihood under the model.

Focusing on BASE models, we find that both generative PPL and self-accuracy improve consistently in the number of replaced tokens (Fig. 3), with a gen. PPL of 93.3 and self-acc. of 73.5% for the $p_u = 0.2$ model (up from 214 and 62.0% respectively). Qualitative evaluation also confirms this (see examples in Tab. 1 and 7). For the mask-only model, while the self-correction step still resamples the same number of tokens, this does not translate to improved gen. PPL or self-accuracy, showing that the ability to self-correct is only acquired if some amount of uniform noise is present during training. Despite this, the mask-only model does appear to improve slightly, which is likely due to numerical limitations: For numerical stability, we actually set $p_u$ to a very small value instead of exactly zero, empirically resulting in $\sim 10$ (out of 262'144) random tokens per batch. Indeed, the MDM (reimpl.) baseline does not exhibit any self-correction abilities at all and in fact makes samples worse during the self-correction step (App. C).

To bolster the point of qualitative improvement, we do LLM-based grading of samples before and after self-correction in terms of clarity, grammaticality, factuality, writing style, and creativity. Significant improvements are observed after self-correction for $p_u > 0$ models in all categories, with the mask-only model showing significant deterioration (Tab. 4). Clarity and grammaticality experience particularly large boosts, which is not surprising given the size and training scale of the model. See Appendix F.1 for prompt and setup details.

## 5.5. Benchmark Performance

Finally, we evaluate our models' language understanding capabilities on a range of benchmarks. Based on the increased difficulty of the hybrid noise setting, we do not expect $p_u > 0$ models to outperform the mask-only case, which is indeed what we find (App. B). Instead, we focus on comparing the best SMALL GIDD+ model to MDM and autoregressive baselines, namely GPT2 (Radford et al., 2019) and a retrained Llama (Touvron et al., 2023a). Our benchmark suite consists of ARC-e and ARC-c (Clark et al., 2018), BoolQ (Clark et al., 2019), Hellaswag (Zellers et al., 2019), PIQA (Bisk et al., 2019), OpenBookQA (Mihaylov et al., 2018), and WinoGrande (Sakaguchi et al., 2019). We find that average accuracy correlates well with validation PPL (Tab. 5). Among diffusion models, the best performing model is GIDD+ ($p_u = 0.0$) trained for only 131B tokens, surpassing models trained for twice as long.[6] While the best diffusion model, GIDD+ ($p_u = 0.0$), outperforms the autoregressive GPT2, the best autoregressive model, Llama (retrain.), still performs best overall. GIDD+ models trained with uniform noise improve with scale but lag behind their mask-only counterparts, which is consistent with their respective valiation PPLs (App. B). This highlights an important difference between *likelihood estimation* (i.e. recognizing realistic samples) and *sample generation* (i.e. creating realistic samples), which do not always correlate perfectly in practice: Despite mask-only models outperforming in likelihood estimation, the picture is flipped when considering their sample quality, indicating that likelihood-based multiple-choice benchmarks may not be enough to holistically evaluate diffusion language models.

## 6. Related Work

Our work builds on a line of discrete diffusion research, with Austin et al. (2023) first introducing the diffusion ELBO to discrete Markov chains, Campbell et al. (2022) extending it to continous time, Lou et al. (2024) deriving an alternative ELBO based on concrete score matching, and concurrent work by Shi et al. (2024), Sahoo et al. (2024), and Ou et al. (2024) proposing a simplified objective for mask-only diffusion. With the exception of Austin et al. (2023) (App. A.2.6), the combination of masking and uniform noise is left unexplored by this line of work. Gu et al. (2022) use this hybrid noise for vector-quantized image generation, but conduct no investigation on the benefits of combining the two noise types. He et al. (2022) propose a noise schedule that degrades different tokens at different rates, depending on their "difficulty" as estimated using BERT, therefore try-

---

[6]While the difference is rather small and can be explained by run-to-run variance, it is possible that the model is overfitting to shorter sequences due to the way we handle long sequences. We use random cropping instead of chunking (App. E), which may over-emphasize short documents in the training corpus.

ing to avoid intermediate mistakes, but stick to mask-only diffusion. Concurrent work has also looked into adaptive denoising orders (Kim et al., 2025) and adaptive loss weights (Ye et al., 2025) as ways to combat the limitations of mask-only diffusion models.

Continuous diffusion has also been adapted to discrete data by doing Gaussian diffusion in an embedding space (Li et al., 2022; Gulrajani & Hashimoto, 2023). Diffusion-like approaches have also been extended to discrete data, with discrete flow matching (Gat et al., 2024) adapting the flow-matching paradigm (Liu et al., 2022; Lipman et al., 2022) and Bayesian flow networks (Graves et al., 2024) adopting the perspective of denoising directly in probability space rather than collapsing the distribution after each step.

Finally, the idea of denoising a combination of masking and uniform noise was popularized by BERT (Devlin et al., 2019), where it was proposed in the context of representation learning.

## 7. Conclusion

We have introduced a new family of generalized interpolating diffusion processes (dubbed GIDD) and successfully applied it in practice. While the extreme scale required to train overall state-of-the-art language models is out of scope for this work, we see great potential in the methods and results described here, but also in diffusion language models more broadly: Self-correction is an area where next-token prediction notoriously has struggled, but as we discovered, this capability comes naturally to diffusion models given the right type of noise. Our work also presents a step towards closing the gap in pure language modeling performance between diffusion and autoregressive models, achieving state-of-the-art perplexity for compute-matched diffusion models thanks to a re-weighted version of our newly proposed GIDD ELBO. Beyond our work, discrete diffusion models respond well to scaling training-time compute like their next-token prediction counterpart, but also provide a natural way to scale test-time compute. By choosing the number of denoising steps, and now also the number of self-correction iterations, one can trade off speed and accuracy depending on the setting. All in all, and given that GIDD opens a design space yet to be explored fully, this may render diffusion language models a promising competitor to autoregressive models in the future.

## Impact Statement

This paper presents work whose goal is to advance the technical state-of-the-art in an area of Machine Learning. It shares potential societal consequences with much of the work in the general area of language modeling and foundation models.

## Acknowledgment

Thank you to Bobby He, Gregor Bachmann, and Tiago Pimentel for their helpful feedback on the writing. Antonio Orvieto and Bernhard Schölkopf acknowledge the financial support of the Hector Foundation.

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

# Contents

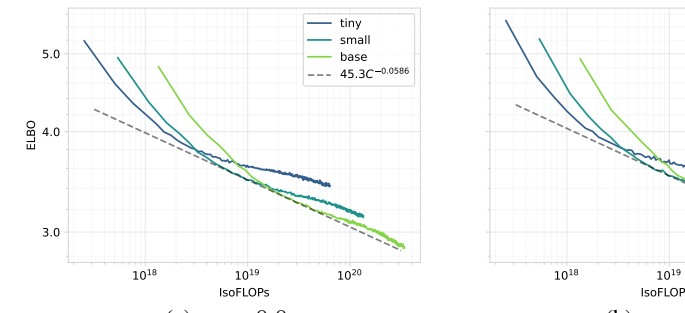 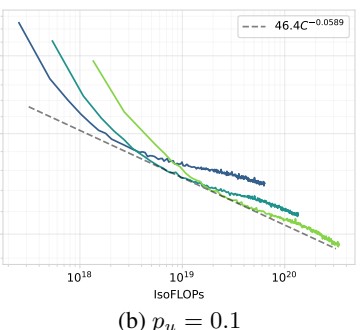 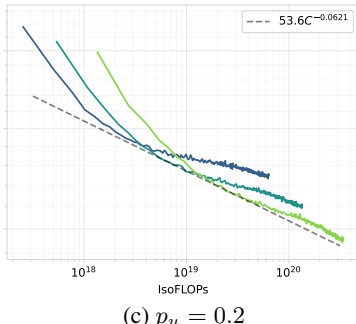

(a) $p_u = 0.0$        (b) $p_u = 0.1$        (c) $p_u = 0.2$

*Figure 4.* Plotting the compute-efficient frontier reveals different scaling behaviors for different uniform noise levels, revealing that training with uniform noise benefits slightly more from scaling compute compared to the mask-only setting.

## A. Uniform Noise and Model Capacity

In Section 5, we have observed that the addition of uniform noise can pose a challenge, and even with improvements to the weighting function, the likelihood of trained models decreases as the proportion of uniform noise increases. Intuitively speaking, this is not entirely surprising since the addition of uniform noise makes the training task strictly more difficult: No longer can the model take for granted that every unmasked token is correct. Instead, it has to consider every token in the context and, if necessary, replace it with the correct one. This intuitive explanation suggests that the reason for the observed discrepancy in performance may be a lack of model capacity, in which case we would expect larger models to be less affected by the addition of uniform noise.

To test this hypothesis, we scale the number of parameters while keeping the training horizon constant and train models of sizes TINY, SMALL, and BASE on different uniform noise levels $p_u \in \{0.0, 0.1, 0.2\}$. We then plot the compute-efficient frontier as an exponential fit to the pareto-optimal validation ELBO (Figure 4). For computing IsoFLOPs of our models, we follow the method from Hoffmann et al. (2022), Appendix F. Due to resource constraints, our setup is somewhat limited: The sample size is limited to three different compute budgets for each noise level, the largest of which is still comparatively small at $3.3 \times 10^{20}$ FLOPs. As a point of reference, many signature capabilities of modern LLMs only start emerging around $10^{22}$ FLOPs (Wei et al., 2022), which is still 2 orders of magnitude higher than our largest compute budget. With this being said, we do indeed observe a consistent albeit small trend of higher levels of uniform noise scaling better with more compute. While the mask-only setting ($p_u = 0.0$) has a scaling exponent of $-0.0586$, adding uniform noise increases the scaling exponent to $-0.0589$ and $-0.0621$ for $p_u = 0.1$ and $p_u = 0.2$ respectively. Extrapolating this trend predicts that the $p_u = 0.2$ setting will overtake $p_u = 0.0$ around $10^{21}$ FLOPs, a compute budget that is routinely reached by mid- to large-scale training runs (Brown et al., 2020; Touvron et al., 2023b; Grattafiori et al., 2024; DeepSeek-AI et al., 2024). However, it has to be stressed that the limitations of our setup make such a prediction highly unreliable. For example, the optimal amount of uniform noise may change with model size and/or compute budget, or certain hyperparameters like the learning rate may have different optimal values depending on $p_u$. Nevertheless, the observed scaling behavior is promising and warrants further investigation.

## B. GIDD Downstream Performance

Benchmark accuracies for GIDD+ models of all three sizes (TINY, SMALL, BASE) and all uniform noise levels ($p_u \in \{0.0, 0.1, 0.2\}$) are given in Table 6. We find that performance improves consistently with model size, regardless of uniform noise level. However, the models trained with uniform noise slightly but consistently lag behind the mask-only model.

## C. Self-Correction Step

**Self-Correction Algorithm.** Our self-correction algorithm is a fixed-point iteration that can be applied to any generated sample that is fully (or partially) denoised. The high-level idea is to query the model to identify tokens that it thinks are wrong and should be replaced, and to then iteratively replace a single token at a time so as to avoid reintroducing conflicting tokens. A pseudocode implementation is given in Algorithm 1. In practice, we find that convergence often comes in the form of oscillation between two or more equally-good states (in terms of self-accuracy), so we additionally implement early-stopping based on self-accuracy. An early-stopping patience of 32 is found to work well.

| Size | Model | Train. toks. | ARC-e | ARC-c | BoolQ | Hellaswag | PIQA | OBQA | WinoG. | Avg. |
|------|-------|-------------|-------|-------|-------|-----------|------|------|--------|------|
| TINY | GIDD+ ($p_u = 0.0$) | 131B | **28.28** | **24.49** | 49.97 | **27.78** | 54.62 | 26.20 | 51.30 | **37.52** |
|  | GIDD+ ($p_u = 0.1$) | 131B | 27.69 | 23.21 | **50.89** | 26.75 | **55.28** | 24.60 | **52.25** | 37.24 |
|  | GIDD+ ($p_u = 0.2$) | 131B | 26.73 | 23.12 | 50.18 | 25.61 | 51.52 | **27.40** | 49.33 | 36.27 |
| SMALL | GIDD+ ($p_u = 0.0$) | 131B | **31.57** | **24.57** | 50.92 | **31.36** | 56.31 | 27.80 | **52.57** | **39.30** |
|  | GIDD+ ($p_u = 0.1$) | 131B | 28.45 | 21.93 | 50.73 | 28.37 | 55.82 | **29.20** | 52.17 | 38.10 |
|  | GIDD+ ($p_u = 0.2$) | 131B | 27.99 | 22.87 | 50.46 | 26.92 | 52.94 | 26.40 | 50.04 | 36.80 |
| BASE | GIDD+ ($p_u = 0.0$) | 131B | **32.58** | **24.40** | 50.86 | **36.62** | 58.05 | 29.2 | 51.54 | **40.46** |
|  | GIDD+ ($p_u = 0.1$) | 131B | 30.13 | 23.04 | **51.10** | 31.91 | 56.15 | 27.6 | **52.33** | 38.89 |
|  | GIDD+ ($p_u = 0.2$) | 131B | 28.75 | 24.15 | 50.95 | 29.82 | 53.81 | 26.8 | 49.25 | 37.65 |

*Table 6.* Downstream performance of GIDD increases consistently with model size, but hybrid noise models lag behind their mask-only counterparts across scales.

---

**Algorithm 1** Self-Correction Step

---

Let $Z_t = (z_t^{(1)}, \ldots z_t^{(L)})$ be a (partially) denoised sequence up to noise level $t$.
Let $f_\theta(Z_t, t)$ denote a (trained) discrete denoising neural network.
**while** not converged **do**
    $\mathbf{x}_\theta^{(1:L)} \leftarrow \text{softmax}(f_\theta(Z_t, t)/\tau)$
    **for** $i \in \{1, \ldots, L\}$ **do**
        $z_t'^{(i)} \sim \text{Cat}(\mathbf{x}_\theta^{(i)})$
    **end for**
    $S \leftarrow \{i | i \in \{1, \ldots, L\} \text{ and } z_t'^{(i)} \neq z_t^{(i)}\}$
    $j \leftarrow \arg\max_{i \in S} \mathbf{x}_\theta^{(i)}(z_t'^{(i)})$
    $z_t^{(j)} \leftarrow z_t'^{(j)}$
**end while**

---

**Additional Results.** In addition to the results for our BASE models given in the main text, we also report improvements for SMALL models, which are very comparable. A notable difference is that for SMALL models, $p_u = 0.1$ has a consistently lower generative PPL, suggesting that $p_u = 0.2$ is too much uniform noise for this size. We also include our MDM baseline, as the comparison between same-sized models is fair. Despite our GIDD+ ($p_u = 0.0$) exhibiting a weak but present ability to self-correct, MDM has no such ability and applying the self-correction step only makes the samples worse (Figure 5). The most likely cause for this difference are implementation details, where, for numerical stability, $p_u$ is not actually set to zero, but to a very small value, which still results in approx. 10 (out of 262'144) random tokens per batch due to limited numerical precision. Alternative explanations may look at differences in hyperparameters, or the numerically non-zero weights on unmasked tokens in the GIDD setup. More examples from the self-correction experiment are given in Table 7.

## D. Number of Denoising Steps

Comparing sample quality for different numbers of denoising steps, we find that, as one would expect, sample quality in terms of generative PPL improves consistently with more denoising steps, up to around 128-256 steps when a plateau is reached (Figure 6). Notably, the sample quality of models trained with uniform noise is significantly better compared to those trained without. This trend is especially strong for small numbers of denoising steps, suggesting that self-correction may help in those scenarios.

## E. Training Details

All our models are based on the DiT architecture (Peebles & Xie, 2023) and use the GPT2 tokenizer (Radford et al., 2019). We train models of three different sizes: TINY ($L = 6$, $H = 8$, $d = 512$; 28.4M non-emb. params.), SMALL ($L = 12$, $H = 12$, $d = 768$; 92.1M non-emb. params.), and BASE ($L = 24$, $H = 16$, $d = 1024$; 321.2M non-emb. params.), where $L$

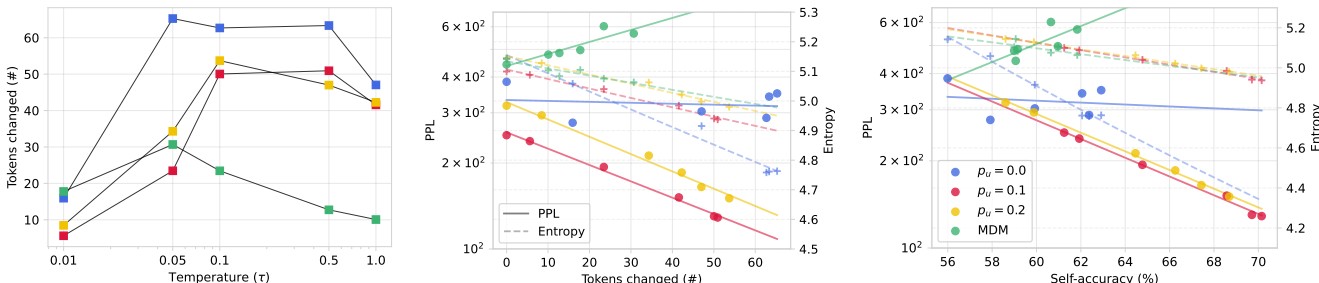

*Figure 5.* Self-correction results for our SMALL models. While the overall trend is the same as for BASE models, the best-performing model uses $p_u = 0.1$ instead of $p_u = 0.2$, suggesting that the ideal uniform noise ratio depends on model size. The MDM baseline is noticeably worse than the mask-only GIDD implementation, with self-correction yielding negative improvements, which is likely due to numerical limitations in the GIDD implementation.

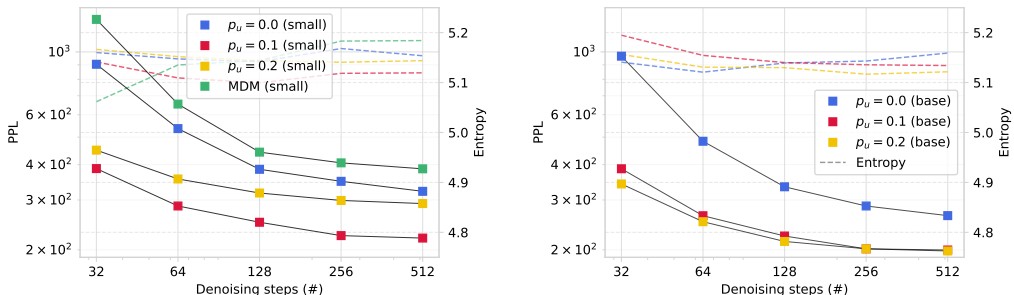

*Figure 6.* Generative PPL (via Gemma 2 9B) decreases monotonically with increasing numbers of denoising steps. Interestingly, the presence of uniform noise during training seems to benefit sample quality overall, but especially so for the low-step regime.

denotes the number of layers, $H$ the number of attention heads, and $d$ the dimensionality of hidden states. All models are trained with a context size of 512 tokens and batch size of 512 for 500k steps (resulting in a total of 131B training tokens) on a single node of 8 NVIDIA A100/H100-80GB GPUs in `bfloat16` precision using Pytorch's mixed precision training (`torch.cuda.autocast`). For the sake of comparison with the literature, some models are trained for twice as long, resulting in 262B training tokens.

For optimization, we use the Adam optimizer (Kingma & Ba, 2017) with $\beta = (0.9, 0.99)$, $\epsilon = 10^{-9}$, and a learning rate of $5 \cdot 10^{-4}$. The learning rate is warmed up linearly for the first 10k steps and then decayed using a cosine schedule to 10% of the initial learning rate. We use weight decay $0.0$ for our ablations (unless stated otherwise) and $0.02$ for the final configuration, also referred to as GIDD+. We also use gradient clipping to a norm of $1.0$.

For our noise schedule, we sample $t \sim \mathcal{U}(\epsilon, 1 - \epsilon)$ with $\epsilon = 10^{-4}$ using low-discrepancy sampling (Kingma et al., 2023). By default, all loss weights (including the unclamped ELBO weighting function) are clipped to $10^4$ to prevent training instability. For sequences longer than 512 tokens we select a random window of 512 tokens, while short sequences are padded to a length of 512. Padding tokens are included in the loss calculation but are ignored in the ELBO.

## F. Evaluation Details

For computing validation metrics, we reserve the last 100k samples ($\approx$1.25%) of the training set (OpenWebText). Validation samples that are longer than the context length are cropped to a random window for consistency with training.

For downstream performance evaluation, we use the `lm-eval-harness`[7] (Gao et al., 2024) with a custom model that uses the ELBO to estimate per-token log-likelihoods. We only consider likelihood-based multiple-choice tasks where the per-token likelihood is computed over both the context and the completion (but not the padding), as preliminary experiments have found that this produces slightly better results. We use $T = 128$ evenly spaced samples (in $[\epsilon, 1 - \epsilon]$) for $t$ to estimate the ELBO. Samples longer than the context size of our model (only applies to BoolQ) are truncated by taking the final $N$

[7]https://github.com/EleutherAI/lm-evaluation-harness

| *Example 1* | |
|---|---|
| Republic of Deltaos have made some significant improvements to the patch in game.2: – "Death in the Vengeance" patch. Notable changes also included: | Republic of Deltaos have made some significant changes to the patch in game v2: The "Death in the Vengeance" patch. Notable changes are included: |
| Proflah Ring can be reached with the ring head up. | Profession Ring can be reached with the ring head up. |
| You can select characters from their application in choice. | You can select characters from their class of choice. |
| Growth returns below your output level when floating between the default dragon and the highest active rev tier. | Growth returns to your output level when floating between the default level and the highest level revamp. |
| Borg followed by Radiant World to earn and the coveted tutorial is now also available for Edition 12. | Borg followed by Radiant World to earn and the coveted tutorial is now also available at level 12. |
| *Example 2* | |
| a new industrial renaissance movement which uses the winner'swould of GE technologies | a new industrial manufacturing platform which uses the lion's share of GE technologies |
| strong link between both US at manufacturing and integral US manufacturing production platform in America | strong link between the US industrial manufacturing and the US industrial manufacturing platform in Europe |
| *Example 3* | |
| short of the feeds public front music ming | instead of the free music fronting service |
| unlimited free music streaming and high-quality content, available whenever you Webs for your subscription. | unlimited free music streaming and high-quality content, available when you pay for a subscription. |
| *Example 4* | |
| Journal publishing has opened the world to these kinds scientists and scientists deserve an encouraging place to look. | Journal publishing has opened the world to these kinds, and scientists have an encouraging way to look. |
| Some researchers can discuss several papers, others are putting many many specific types of material. | Some researchers openly discuss their papers, others are putting many many specific types of papers. |
| Globality postulates the circumstances – researchers learn from the very reputation of other study team. | Globality postulates the circumstances – researchers learn from the good work of other research team. |

*Table 7.* Examples from our self-correction experiments reveal a noticeable qualitative improvement: The model is able to correct grammatical mistakes (Ex. 2, 3), improve coherence (Ex. 3), and improve the choice of words given the context (Ex. 1, 4). The examples are from GIDD+ BASE ($p_u = 0.2$) with self-correction temperature $\tau = 0.1$.

```
1. Clarity and coherence: Keeping in mind that the text may be cut off in the beginning
and at the end due to it being an excerpt, how clear and understandable is the text?
2. Grammaticality: Are there any grammatical errors in the text?
3. Factuality: If applicable, is the factually verifiable information stated in the text
(e.g. facts about geography, history, etc.) accurate and reliable?
4. Writing style: How well is the text written in terms of style and fluency? Do the
sentences flow well, is the vocabulary appropriate?
5. Creativity: How original and creative is the text?

For each category, give a short justification before providing the final score. Your
answer should be following the JSON format, with one top-level key for each aspect ('
clarity', 'grammaticality', 'factuality', 'style', and 'creativity').
Each aspect, in turn, should be a JSON object consisting of a 'reasoning' and 'score' key
 in that order. The 'reasoning' key should contain a short justification for the score,
and the 'score' key should contain the score itself.

Please keep the following in mind:
- Give your justification first before deciding on a final score.
- Only output the JSON containing the justifications and scores and nothing else.
- Keep in mind that the presented paragraph may be an excerpt from a longer document, so
it may not be fully self-contained. Do not deduct points for issues arising from this.

The text to be graded is as follows:
```
{text}
```
```

*Figure 7.* Prompt used for the LLM-based evaluation of sample quality.

tokens, similar to context scrolling for autoregressive models.

Our generative perplexity is based on the `google/gemma-2-9b`[8] model (Gemma Team, 2024) as it provides a good tradeoff between language modeling accuracy and efficiency. Prior work often relies on the GPT2-large model for generative PPL computation, but we believe that in order to draw meaningful conclusions, it is crucial to use a grading model that is sufficiently more capable than the graded model in order reasonably provide a proxy of the "ground truth" distribution of natural language.

Unigram entropy is computed following Zheng et al. (2025) (App. H.1) by computing the entropy of the token-level frequency distribution over unique tokens in the sequence. This means that, for our maximum sequence length of 512, the upper bound is $\log(512) \approx 6.24$ in case all tokens are unique.

We use the GNU `parallel` software (Tange, 2024) for streamlining the execution of our evaluation scripts.

### F.1. LLM-based Evaluation

We qualitatively evaluate unconditionally generated samples before and after self-correction using the GPT-4o API (`gpt-4o-2024-08-06`) by instructing it to grade the samples in terms of clarity, grammaticality, factuality, writing style, and creativity on a scale from 1–10. The model is provided a sample text and instructed to first give a justification and then a grade for each category, and to return the result as a JSON string for ease of parsing. See Figure 7 for the exact prompt used.

## G. Evaluating Generative Perplexity of Diffusion Models

Generative perplexity is an evaluation metric intended to measure the quality of generated samples with a grading model that is used as a proxy of the ground truth data distribution. Under this assumption, we deem samples with a high likelihood under the grading model to be of higher quality or at least to be more likely under the data distribution. While there are many potential issues with this approach, results can be particularly misleading if the grading model is a bad proxy for the

---

[8]https://huggingface.co/google/gemma-2-9b

| Min-$p$ | Sample |
|---|---|
| | **Model: GIDD+ (SMALL, $p_u = 0.0$)** |
| $p = 10^{-7}$ | The second time in a month Henrik Zqvist's wrist is nothing concerning for Perproductu, the American.\n\n"It's going to make a break," Zetterberg said, "And hope it is comfortable enough for the next three games. I got to practice and we had warm runs this week."\n\nCavoring? " for a handful of games."\n\nPlacing \$6.9 million for the 2016 2013 first-round pick, the Americans are happy to have his availability now and offered him some flexibility.\n\nThe Wings' expectations are varied in both player nature and rotation.\n\n"It's a different situation as we know I'm going to be around a little bit more (t my right wrist), [...] |
| $p = 10^{-5}$ | The majority of high-risk projects appear to be delayed indefinitely because another is sought to replace them.\n\nIt is understood that the council has set up a commission to decide the timeframe of what are expected to be by Christmas, starting on 15 October.\n\n"We have undertaken a thorough and rather robust assessment of the ongoing activity in the council, so it is considered that the period to justify a review is further too long," reads a submission to councillors.\n\nThe review is being carried out, 10 months after the Conservative administration lodged an election proposal in April.[...] |
| $p = 10^{-3}$ | \n\n6\n\n2\n\n3\n\n4\n\n3\n\n6\n\n5\n\n8\n\n1\n\n4\n\n8\n\n\n8\n\n\n\n8\n\n\n1\n\n\n\n5\n\n5\n\n\n4\n\n2\n\n2\n\n1\n\n\n2\n\n8\n\n\n4\n\n\n\n2\n\n2\n\n8\n\n2\n\n\n2\n\n\n4\n\n2\n\n\n\n\n2\n\n8\n\n\n\n\n1\n\n4\n\n1\n\n2\n\n4\n\n4\n\n2\n\n\n\n\n\n2\n\n\n\n1\n\n1\n\n1\n\n1\n\n1\n\n2\n\n2\n\n\n\n\n3\n\n2\n\n\n\n\n\n1\n\n2\n\n\n\n\n1\n\n\n\n\n\n2\n\n3\n\n4\n\n3\n\n4\n\n\n\n\n2\n\n2\n\n5\n\n\n\n\n\n\n\n1\n\n\n\n\n\n2\n\n\n\n\n2\n\n3\n\n4\n2\n\n52\n\n2[...] |
| | **Model: MDM (SMALL, reimpl.)** |
| $p = 10^{-7}$ | Incredible. David Segol making his ankle...... (Photo by photo) (quotes from Evan Jones) ...\n\nThe Nigerian was wearing tight pants since being 'anemic'.\n\n"I basically wanted French au shorts in one piece to wear with everything that was on Internet in 2010," he said, wearing Section 50 Lo Lim Channt out. "The things w that I wear things that I have to work."\n\nHe is using a words, which means "Planned for the swollen collarbone," he is also team by fall good treatment. Ponder Mikko the kn/sw writer that during one World Leberbiroux has since made his lap![...] |
| $p = 10^{-5}$ | Crowwick said the staff has a "been growing" around the Academy of Newcastle City\n\nThe club's Academy which works with youth players is being used in the first time as the first look teams on their new training starting ground.\n\nThe Newcastle National School will grow players, develop coaches and implement the system for the rest of the game.\n\nCoach Steve Curtis said the club has been keen to show players that the youngsters can develop the new system at international level.\n\n"We're hoping those who come that will learn his way there and [...] |
| $p = 10^{-3}$ | A vehicle is in the side of vehicle of, located on the side of the car by Facebook.\n\nA car is located on the rear of the car of, in side of area.\n\nThe driver of the vehicle is in the area of the rear of, located on the side of the vehicle from Facebook.com.\n\nA driver has parked the area of the vehicle on the side, side of the Road of.\n\nThe owner of the vehicle has been located on the side of the rear of, located on the side of a road on the road of[?] of, the Newark, N.J.\n\nThe owner is attempting to locate passenger in the area of vehicle and parked [...] |

*Table 8.* Samples generated with different min-$p$ cutoff values. Despite generative PPL consistently decreasing for larger cutoffs, the sample quality starts to deteriorate drastically for values larger than $p \geq 10^{-4}$.

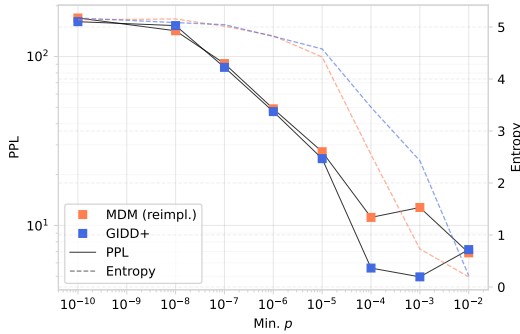

*Figure 8.* Generative perplexity as measured by GPT2-large decreases consistently as the min-$p$ cutoff is increased. Unfortunately, this trend does not correlate with subjective quality, showing a major limitation of generative PPL with GPT2-large.

true likelihood of samples. While prior work often relies on GPT2-large as a grading model, we find that this model suffers from failure modes that are typical for small (by today's standards) models.

Before going into detail about the exact failure modes we observe, we need to discuss another peculiarity that arises when sampling from discrete diffusion models. A common approach to efficiently sample a categorical distribution is the Gumbel-max trick (Gumbel, 1954), which is used by Shi et al. (2024) (via JAX categorical sampling) and Sahoo et al. (2024). As noted by Shi et al. (2024) in App. G, this approach is somewhat numerically unstable and leads to an implicit regularization of the sampling distribution, effectively masking out very small probabilities (the authors report smaller than $10^{-8}$). This is effectively a min-$p$ sampling adapter, which has been found to improve sample quality of autoregressive language models as well (Nguyen et al., 2024). To study the effective of this min-$p$ regularization, we explicitly implement it with a more numerically stable sampling algorithm based on binary search. Measuring the generative PPL as per GPT2-large for different values of $p$, we find a consistent decrease with the observed gen. PPL of $\sim$90 around $p = 10^{-7}$ being consistent with what is reported for SMALL models in the literature (Fig. 8).

However, as the cutoff probability increases, generative PPL drops to suspiciously low values. Indeed, manual inspection of the samples reveals that the sample quality deteriorates drastically for larger cutoffs (examples in Tab. 8). These samples exhibit low diversity and a lot of repetition, failure modes that are typical for small (autoregressive) language models. Despite this, these samples have very high likelihood under GPT2-large and hence are considered "high quality" under the generative PPL metric. This shows the importance of using more capable grading models that are able to pick up on these failure modes and the effect that sampling parameters can have on the outcome of these experiments. It also highlights the limitations of comparing absolute generative PPL numbers, as they can be misleading in isolation. This is why unigram entropy can help spot and quantify any catastrophic collapse in diversity, though it also struggles to quantify more subtle losses of diversity in the min-$p \leq 10^{-6}$ regime (Fig. 8). While Gemma 2 9B also suffers from the failure mode described here to some extent, it should provide a much better proxy of the "true distribution" of natural language compared to the much weaker GPT2-large.

# H. Proofs

## H.1. Conditional Mixing Schedule

**Lemma H.1.** *If $\alpha_{t|s}$ and $\beta_{t|s}\boldsymbol{\pi}_{t|s}$ are defined as in Proposition 3.3, and if $\beta_{t|s} = \beta_t - \frac{\alpha_t}{\alpha_s}\beta_s$, then $\alpha_{t|s} + \beta_{t|s} = 1$ and $\boldsymbol{\pi}_{t|s}$ is a prob. vector, i.e. $\boldsymbol{\pi}_{t|s} > 0$ and $\mathbf{1}^\top \boldsymbol{\pi}_{t|s} = 1$.*

*Proof.* We have

$$\alpha_{t|s} + \beta_{t|s} = \frac{\alpha_t}{\alpha_s} + \beta_t - \frac{\alpha_t}{\alpha_s}\beta_s = \frac{\alpha_t}{\alpha_s}(1 - \beta_s) + \beta_t = \frac{\alpha_t}{\alpha_s}\alpha_s + \beta_t = \alpha_t + \beta_t = 1 \tag{21}$$

and, using the fact that $\mathbf{1}^\top \boldsymbol{\pi}_t = 1, \forall t$ by Definition 3.2,

$$\mathbf{1}^\top \boldsymbol{\pi}_{t|s} = \mathbf{1}^\top \frac{1}{\beta_{t|s}}(\beta_{t|s}\boldsymbol{\pi}_{t|s}) = \frac{1}{\beta_{t|s}}\left(\beta_t \mathbf{1}^\top \boldsymbol{\pi}_t - \frac{\alpha_t}{\alpha_s}\beta_s \mathbf{1}^\top \boldsymbol{\pi}_s\right) = \frac{1}{\beta_{t|s}}\left(\beta_t - \frac{\alpha_t}{\alpha_s}\beta_s\right) = 1, \tag{22}$$

thus proving the claim. □

## H.2. GIDD Conditional Transitions

*Proof of Telescoping in Prop. 3.3.* Recall the recursive formulas for $\alpha_{t|s}$ and $\beta_{t|s}\boldsymbol{\pi}_{t|s}$:

$$\alpha_{t+\Delta|s} = \dot{\alpha}_t \alpha_{t|s} \tag{23}$$

$$\beta_{t+\Delta|s}\boldsymbol{\pi}_{t|s} = \dot{\beta}_t \dot{\boldsymbol{\pi}}_t + \dot{\alpha}_t \beta_{t|s}\boldsymbol{\pi}_{t|s} \tag{24}$$

By unrolling the recursion and plugging in the definition of $\dot{\alpha}_t$ we then have

$$\alpha_{t|s} = \dot{\alpha}_{t-\Delta}\dot{\alpha}_{t-2\Delta}\cdots\dot{\alpha}_s \underbrace{\alpha_{s|s}}_{=1} = \prod_{i=s/\Delta}^{t/\Delta-1}\dot{\alpha}_{\Delta i} = \prod_{i=s/\Delta}^{t/\Delta-1}\frac{\alpha_{\Delta(i+1)}}{\alpha_{\Delta i}} = \frac{\alpha_{\Delta(t/\Delta)}}{\alpha_{\Delta(s/\Delta)}} = \frac{\alpha_t}{\alpha_s}. \tag{25}$$

Analogously, for $\beta_{t|s}\boldsymbol{\pi}_{t|s}$ we get

$$\beta_{t|s}\boldsymbol{\pi}_{t|s} =$$

$$\dot{\beta}_{t-\Delta}\dot{\boldsymbol{\pi}}_{t-\Delta} + \dot{\alpha}_{t-\Delta}\dot{\beta}_{t-2\Delta}\dot{\boldsymbol{\pi}}_{t-2\Delta} + \cdots + \dot{\alpha}_{t-\Delta}\cdots\dot{\alpha}_{s+\Delta}\dot{\beta}_s\dot{\boldsymbol{\pi}}_s + \dot{\alpha}_{t-\Delta}\cdots\dot{\alpha}_s \underbrace{\beta_{s|s}\boldsymbol{\pi}_{s|s}}_{=0} = \sum_{i=s/\Delta}^{t/\Delta-1}\alpha_{t|\Delta(i+1)}\dot{\beta}_{\Delta i}\dot{\boldsymbol{\pi}}_{\Delta i}$$

$$= \sum_{i=s/\Delta}^{t/\Delta-1}\frac{\alpha_t}{\alpha_{\Delta(i+1)}}\left(\beta_{\Delta(i+1)}\boldsymbol{\pi}_{\Delta(i+1)} - \frac{\alpha_{\Delta(i+1)}}{\alpha_{\Delta i}}\beta_{\Delta i}\boldsymbol{\pi}_{\Delta i}\right)$$

$$= \sum_{i=s/\Delta}^{t/\Delta-1}\left(\frac{\alpha_t}{\alpha_{\Delta(i+1)}}\beta_{\Delta(i+1)}\boldsymbol{\pi}_{\Delta(i+1)} - \frac{\alpha_t}{\alpha_{\Delta i}}\beta_{\Delta i}\boldsymbol{\pi}_{\Delta i}\right)$$

$$= \frac{\alpha_t}{\alpha_t}\beta_t\boldsymbol{\pi}_t - \frac{\alpha_t}{\alpha_s}\beta_s\boldsymbol{\pi}_s = \beta_t\boldsymbol{\pi}_t - \frac{\alpha_t}{\alpha_s}\beta_s\boldsymbol{\pi}_s, \tag{26}$$

yielding the desired expressions for $\alpha_{t|s}$ and $\beta_{t|s}\boldsymbol{\pi}_{t|s}$ and concluding the proof. □

## H.3. GIDD Forward Rate

*Proof of Lemma 3.6.* We need to show that the CTMC forward rate matrix $R_t$ of GIDD is given by

$$R_t(z_s, z_t) = \frac{\alpha'_t}{\alpha_t}\delta_{z_s,z_t} + \mathbf{z}_t^\top\left(\beta_t\boldsymbol{\pi}'_t - \frac{\alpha'_t}{\alpha_t}\boldsymbol{\pi}_t\right), \tag{27}$$

where $\alpha'_t$ and $\boldsymbol{\pi}'_t$ denote the time-derivative of the respective mixing function.

The proof follows the idea of Proof 2 in Campbell et al. (2022), App. B.2 to perform a first-order Taylor expansion on $q_{t|s}(z_t|z_s)$. To this end, let $s$ be given and let $t = s + \Delta$ for some positive $\Delta \to 0$. Then, by Proposition 3.3 we have

$$q_{s+\Delta|s}(z_t|z_s) = \mathbf{z}_t^\top Q_{s+\Delta|s}\mathbf{z}_s = \mathbf{z}_t^\top(\alpha_{s+\Delta|s}\mathbf{z}_s + \beta_{s+\Delta|s}\boldsymbol{\pi}_{s+\Delta|s}). \tag{28}$$

We now linearize $\alpha_{s+\Delta|s}$ and $\beta_{s+\Delta|s}\boldsymbol{\pi}_{s+\Delta|s}$ around $s$, resulting in

$$\alpha_{s+\Delta|s} = \frac{\alpha_{s+\Delta}}{\alpha_s} = \frac{\alpha_s + \alpha'_s\Delta + o(\Delta)}{\alpha_s} = 1 + \frac{\alpha'_s}{\alpha_s}\Delta + o(\Delta), \tag{29}$$

$$\beta_{s+\Delta|s}\boldsymbol{\pi}_{s+\Delta|s} = \beta_{s+\Delta}\boldsymbol{\pi}_{s+\Delta} - \frac{\alpha_{s+\Delta}}{\alpha_s}\beta_s\boldsymbol{\pi}_s$$

$$= (\beta_s\boldsymbol{\pi}_s + (\beta_s\boldsymbol{\pi}_s)'\Delta + o(\Delta)) - \left(1 + \frac{\alpha'_s}{\alpha_s}\Delta + o(\Delta)\right)\beta_s\boldsymbol{\pi}_s \tag{30}$$

$$= \left((\beta_s\boldsymbol{\pi}_s)' - \frac{\alpha'_s}{\alpha_s}\beta_s\boldsymbol{\pi}_s\right)\Delta + o(\Delta).$$

The inner term of Eq. 30 can be simplified as follows using the product rule:

$$(\beta_s \boldsymbol{\pi}_s)' - \frac{\alpha_s'}{\alpha_s}\beta_s \boldsymbol{\pi}_s = (1 - \alpha_s)' \boldsymbol{\pi}_s + \beta_s \boldsymbol{\pi}_s' - \frac{\alpha_s'}{\alpha_s}(1 - \alpha_s)\boldsymbol{\pi}_s \tag{31a}$$

$$= \beta_s \boldsymbol{\pi}_s' - \alpha_s'\left(1 + \frac{1 - \alpha_s}{\alpha_s}\right)\boldsymbol{\pi}_s \tag{31b}$$

$$= \beta_s \boldsymbol{\pi}_s' - \frac{\alpha_s'}{\alpha_s}\boldsymbol{\pi}_s \tag{31c}$$

Plugging this into $q_{s+\Delta}(z_t|z_s)$ yields

$$q_{s+\Delta|s}(z_t|z_s) = \mathbf{z}_t^\top \left(\left(1 + \frac{\alpha_s'}{\alpha_s}\Delta + o(\Delta)\right)\mathbf{z}_s + \left(\beta_s \boldsymbol{\pi}_s' - \frac{\alpha_s'}{\alpha_s}\boldsymbol{\pi}_s\right)\Delta + o(\Delta)\right)$$

$$= \delta_{z_s,z_t} + \underbrace{\left(\frac{\alpha_s'}{\alpha_s}\delta_{z_s,z_t} + \mathbf{z}_t^\top\left(\beta_s \boldsymbol{\pi}_s' - \frac{\alpha_s'}{\alpha_s}\boldsymbol{\pi}_s\right)\right)}_{=R_s(z_s,z_t)}\Delta + o(\Delta), \tag{32}$$

which presents the rate matrix as claimed, concluding the proof. $\qquad\square$

## H.4. GIDD Backward Rate

**Definition H.2** (CTMC Backward Transition). For any $s < t$ with $s = t - \Delta$ and $\Delta \to 0$, we have

$$p_{s|t}(z_s|z_t) = \delta_{z_t,z_s} + \hat{R}_t(z_t, z_s) + o(\Delta), \tag{33}$$

where $\hat{R}_t$ is called the backward transition rate.

**Lemma H.3** (GIDD Backward Rate). *The CTMC backward rate matrix $\hat{R}_t^\theta$ of $p_\theta(z_s|z_t)$ as defined in Equation 12 is given by*

$$\hat{R}_t^\theta(z_t, z_s) = -\delta_{z_s,z_t}\sum_{z'} R_t(z', z_t)\frac{q_t(z'|\mathbf{x}_\theta)}{q_t(z_t|\mathbf{x}_\theta)} + R_t(z_s, z_t)\frac{q_t(z_s|\mathbf{x}_\theta)}{q_t(z_t|\mathbf{x}_\theta)}. \tag{34}$$

*Proof.* For the reverse rate matrix $\hat{R}_t^\theta(z_t, z_s)$, we start with our choice for the model backward transition (Eq. 12):

$$p_\theta(z_s|z_t) = q_{t|s}(z_t|z_s)\frac{q_s(z_s|\mathbf{x}_\theta)}{q_t(z_t|\mathbf{x}_\theta)} \tag{35}$$

By now setting $s = t - \Delta$ with $\Delta \to 0$, we get

$$p_\theta(z_s|z_t) = q_{t|t-\Delta}(z_t|z_s)\frac{q_{t-\Delta}(z_s|\mathbf{x}_\theta)}{q_t(z_t|\mathbf{x}_\theta)} \tag{36a}$$

$$= (\delta_{z_t,z_s} + R_{t-\Delta}(z_s, z_t)\Delta + o(\Delta))\frac{q_{t-\Delta}(z_s|\mathbf{x}_\theta) - q_{t-\Delta}'(z_s|\mathbf{x}_\theta)\Delta + o(\Delta)}{q_t(z_t|\mathbf{x}_\theta)} \tag{36b}$$

$$\stackrel{\Delta \to 0}{=} \underbrace{\delta_{z_s,z_t}\frac{q_t(z_s|\mathbf{x}_\theta)}{q_t(z_t|\mathbf{x}_\theta)}}_{=1 \text{ if } z_s = z_t} + \underbrace{\left(\delta_{z_s,z_t}\frac{-q_t'(z_s|\mathbf{x}_\theta)}{q_t(z_t|\mathbf{x}_\theta)} + R_t(z_s, z_t)\frac{q_t(z_s|\mathbf{x}_\theta)}{q_t(z_t|\mathbf{x}_\theta)}\right)}_{=\hat{R}_t^\theta(z_t,z_s)}\Delta + o(\Delta) \tag{36c}$$

In order to get rid of the time-derivative of the forward process we use Kolmogorov's forward equation to rewrite the time-derivative of the forward process $q_t'(z_s|\mathbf{x}_\theta)$ as $q_t'(z_s|\mathbf{x}_\theta) = \sum_{z'} q_t(z'|\mathbf{x}_\theta)R_t(z', z_s)$, resulting in

$$\hat{R}_t^\theta(z_t, z_s) = -\delta_{z_s,z_t}\sum_{z'}\frac{q_t(z'|\mathbf{x}_\theta)}{q_t(z_t|\mathbf{x}_\theta)}R_t(z', z_s) + R_t(z_s, z_t)\frac{q_t(z_s|\mathbf{x}_\theta)}{q_t(z_t|\mathbf{x}_\theta)}. \tag{37}$$

Finally, we rename $z_s$ to $z_t$ in the first term, since they are equal if $\delta_{z_s,z_t} = 1$ and the entire term is 0 otherwise, resulting in the desired equality:

$$\hat{R}_t^\theta(z_t, z_s) = -\delta_{z_s,z_t}\sum_{z'} R_t(z', z_t)\frac{q_t(z'|\mathbf{x}_\theta)}{q_t(z_t|\mathbf{x}_\theta)} + R_t(z_s, z_t)\frac{q_t(z_s|\mathbf{x}_\theta)}{q_t(z_t|\mathbf{x}_\theta)}. \tag{38}$$

$\qquad\square$

## H.5. GIDD ELBO

Our starting point is a slightly modified version of the continuous-time ELBO (CT-ELBO) from Campbell et al. (2022) which explicitly keeps some constant terms. Keeping these terms is useful for canceling out other terms in subsequent derivations. The proof of Proposition H.4 is largely analogous to Campbell et al. (2022).

**Proposition H.4.** *For any CTMC diffusion process with marginals $q_t(z_t|x)$, forward rate $R_t(z_s, z_t)$, and backward rate $\hat{R}_t(z_s, z_t)$, the CT-ELBO is given by*

$$\log p(x) \geq \mathbb{E}_{t,q_t(z_t|x)}\left[\hat{R}_t(z_t, z_t) - R_t(z_t, z_t) + \sum_{z' \neq z_t} R_t(z', z_t)\frac{q_t(z'|x)}{q_t(z_t|x)} \log \frac{\hat{R}_t(z_t, z')q_t(z_t|x)}{R_t(z', z_t)q_t(z'|x)}\right] + C, \tag{39}$$

*where $t \sim \mathcal{U}(0, 1)$ and $C = \mathbb{E}_{q_0(z_0|x)}[\log p(x|z_0)] - D_{KL}(q_1(z_1|x)\|p_1(x_1))$.*

*Proof.* The key quantity in the diffusion ELBO is the KL-divergence between the true and the model backward transition, which is given by the following. For the second equality, we use Baye's rule and the fact that $q_{t|s}(z_t|z_s, x) = q_{t|s}(z_t|z_s)$ following the Markovian property of the forward process:

$$\begin{aligned}
D_{KL}(q_{s|t}(z_s|z_t, x)\|p_{s|t}(z_s|z_t)) &= \sum_{z_s} q_{s|t}(z_s|z_t, x) \log \frac{q_{s|t}(z_s|z_t, x)}{p_{s|t}(z_s|z_t)} \\
&= \sum_{z_s} q_{t|s}(z_t|z_s)\frac{q_s(z_s|x)}{q_t(z_t|x)} \log \frac{q_{t|s}(z_t|z_s)q_s(z_s|x)}{p_{s|t}(z_s|z_t)q_t(z_t|x)}
\end{aligned} \tag{40}$$

In order to derive the continuous-time ELBO, we first analyze the behaviour of this term as $\Delta \to 0$ with $s = t - \Delta$. By the definition of CTMC, we then have

$$\begin{aligned}
\log q_{t|s}(z_t|z_s) &= \log(\delta_{z_t, z_s} + \Delta R_s(z_s, z_t) + o(\Delta)) \\
&= \delta_{z_t, z_s}\Delta R_s(z_s, z_t) + (1 - \delta_{z_t, z_s}) \log(\Delta R_s(z_s, z_t) + o(\Delta)) + o(\Delta),
\end{aligned} \tag{41}$$

where we use the fact that $\log(1 + x) = x - \frac{x^2}{2} + o(x^2)$ for the $z_s = z_t$ case. By also using the fact that

$$\Delta \log(\Delta x + o(\Delta)) = \Delta \log \Delta + \Delta \log(x + o(1)) = \Delta \log \Delta + \Delta \log(1 + o(1)) + \Delta \log x \stackrel{\Delta \to 0}{=} \Delta \log x, \tag{42}$$

we then get that

$$\begin{aligned}
q_{t|s}(z_t|z_s) \log q_{t|s}(z_t|z_s) &= [\delta_{z_t, z_s} + \Delta R_s(z_s, z_t) + o(\Delta)] \\
&\quad \cdot [\delta_{z_t, z_s}\Delta R_s(z_s, z_t) + (1 - \delta_{z_t, z_s}) \log(\Delta R_s(z_s, z_t) + o(\Delta)) + o(\Delta)] \\
&= \delta_{z_t, z_s}\Delta R_s(z_s, z_t) + \underbrace{\delta_{z_t, z_s}(1 - \delta_{z_t, z_s})(\dots)}_{=0} + (1 - \delta_{z_t, z_s})R_s(z_s, z_t)\underbrace{\Delta \log(\Delta R_s(z_s, z_t) + o(\Delta))}_{=\Delta \log R_s(z_s, z_t)} + o(\Delta) \\
&= \delta_{z_t, z_s}\Delta R_s(z_s, z_t) + (1 - \delta_{z_t, z_s})\Delta R_s(z_s, z_t) \log R_s(z_s, z_t) + o(\Delta).
\end{aligned} \tag{43}$$

By analogous reasoning, we also get that

$$q_{t|s}(z_t|z_s) \log p_{s|t}(z_s|z_t) = \delta_{z_t, z_s}\Delta \hat{R}_s + (1 - \delta_{z_t, z_s})\Delta R_s(z_s, z_t) \log \hat{R}_s(z_t, z_s) + o(\Delta). \tag{44}$$

Finally, we also use the fact that

$$\begin{aligned}
q_{t|s}(z_t|z_s) \underbrace{\log \frac{q_s(z_s|x)}{q_t(z_t|x)}}_{=0 \text{ if } z_s = z_t} &= (1 - \delta_{z_t, z_s})(\delta_{z_t, z_s} + \Delta R_s(z_s, z_t) + o(\Delta)) \log \frac{q_s(z_s|x)}{q_t(z_t|x)} \\
&= (1 - \delta_{z_t, z_s})\Delta R_s(z_s, z_t) \log \frac{q_s(z_s|x)}{q_t(z_t|x)} + o(\Delta).
\end{aligned} \tag{45}$$

Now we plug Equations 43, 44, and 45 into Equation 40, which yields

$$D_{KL}(q_{s|t}(z_s|z_t,x)\|p_{s|t}(z_s|z_t)) = \sum_{z_s} q_{t|s}(z_t|z_s) \frac{q_s(z_s|x)}{q_t(z_t|x)} \log \frac{q_{t|s}(z_t|z_s)q_s(z_s|x)}{p_{s|t}(z_s|z_t)q_t(z_t|x)}$$

$$= \sum_{z_s} \frac{q_s(z_s|x)}{q_t(z_t|x)} \left( \underbrace{q_{t|s}(z_t|z_s) \log q_{t|s}(z_t|z_s)}_{\text{Eq. 42}} - \underbrace{q_{t|s}(z_t|z_s) \log p_\theta(z_t|z_s)}_{\text{Eq. 43}} + \underbrace{q_{t|s}(z_t|z_s) \frac{q_s(z_s|x)}{q_t(z_t|x)}}_{\text{Eq. 44}} \right)$$

$$= \underbrace{\Delta R_s(z_t, z_t) - \Delta \hat{R}_s(z_t, z_t)}_{\text{if } z_s = z_t} + \sum_{z_s \neq z_t} \Delta R_s(z_s, z_t) \log \frac{R_s(z_s, z_t)q_s(z_s|x)}{\hat{R}_s(z_t, z_s)q_t(z_t|x)} + o(\Delta)$$

$$= \Delta \left( R_s(z_t, z_t) - \hat{R}_s(z_t, z_t) + \sum_{z_s \neq z_t} R_s(z_s, z_t) \log \frac{R_s(z_s, z_t)q_s(z_s|x)}{\hat{R}_s(z_t, z_s)q_t(z_t|x)} \right) + o(\Delta). \quad (46)$$

We now substitute this result into the discrete-time diffusion ELBO, which is given by

$$\log p(x) \geq -\sum_{i=2}^{T} \mathbb{E}_{q_{\Delta i}(z_t|x)} \left[ D_{KL}(q_{\Delta(i-1)|\Delta i}(z_s|z_t,x)\|p_\theta(z_s|z_t)) \right] + C$$

$$= -(T-2)\mathbb{E}_{t\sim\mathcal{U}\{2\Delta,3\Delta,\dots,(T-1)\Delta,1\},q_t(z_t,x)} \left[ D_{KL}(q_{t-\Delta|t}(z_s|z_t,x)\|p_\theta(z_s|z_t)) \right] + C, \quad (47)$$

where $C$ is the standard diffusion ELBO constant with $C = \mathbb{E}_{q_0(z_0|x)}[\log p(x|z_0)] - D_{KL}(q_1(z_1|x)\|p(x_1))$. Substituting and taking $\Delta = \frac{1}{T} \to 0$ results in the final CT-ELBO:

$$\log p(x) \geq -\sum_{i=2}^{T} \mathbb{E}_{q_{i\Delta}(z_t|x)} \left[ D_{KL}(q_{(i-1)\Delta|i\Delta}(z_s|z_t,x)\|p_\theta(z_s|z_t)) \right] + C$$

$$\overset{s=t-\Delta}{=} - \underbrace{(1/\Delta - 2)}_{\to 1/\Delta \text{ as } \Delta \to 0} \mathbb{E}_{t\sim\mathcal{U}\{2\Delta,3\Delta,\dots,(T-1)\Delta,1\},q_t(z_t,x)} \left[ D_{KL}(q_{s|t}(z_s|z_t,x)\|p_\theta(z_s|z_t)) \right] + C$$

$$\overset{\Delta\to 0}{=} -\frac{1}{\Delta} \mathbb{E}_{t\sim\mathcal{U}(0,1),q(z_t|x)} \left[ \Delta \left( R_t(z_t, z_t) - \hat{R}_t(z_t, z_t) + \sum_{z_s \neq z_t} R_t(z_s, z_t) \log \frac{R_t(z_s, z_t)q_t(z_s|x)}{\hat{R}_t(z_t, z_t)q_t(z_t|x)} \right) + o(\Delta) \right] + C$$

$$= \mathbb{E}_{t\sim\mathcal{U}(0,1),q_t(z_t|x)} \left[ \hat{R}_t(z_t, z_t) - R_t(z_t, z_t) + \sum_{z' \neq z_t} R_t(z', z_t) \log \frac{\hat{R}_t(z_t, z')q_t(z_t|x)}{R_t(z', z_t)q_t(z'|x)} \right] - \underbrace{\frac{o(\Delta)}{\Delta}}_{\to 0} + C, \quad (48)$$

which is the desired expression, concluding the proof. $\qquad\square$

Starting at this general form, we now plug in the GIDD forward and backward rates $R_t(z_s, z_t)$ and $\hat{R}_t^\theta(z_t, z_s)$ into Proposition H.4 and simplify the resulting expression to derive the ELBO for GIDD.

*Proof of Theorem 3.7 (GIDD ELBO).* We need to show that

$$-\log p(x) \leq \mathbb{E}_{t,z_t} \left[ w_t(z_t, x) \left( D_{KL}(q_t(\cdot|x)\|q_t(\cdot|\mathbf{x}_\theta)) + D_{IS}(q_t(z_t|x)\|q_t(z_t|\mathbf{x}_\theta)) \right) \right] + C, \quad (49)$$

with $D_{IS}(p\|q) = p/q - \log p/q - 1$, $t \sim \mathcal{U}(0,1)$, $z_t \sim q_t(\cdot|x)$, $C = \mathbb{E}_{q_0(z_0|x)}[\log p(x|z_0)] - D_{KL}(q_1(z_1|x)\|p_1(x_1))$, and the weighting function

$$w_t(z_t, x) = \frac{1}{q_t(z_t|x)} \mathbf{z}_t^\top \left( \beta_t \boldsymbol{\pi}_t' - \frac{\alpha_t'}{\alpha_t} \boldsymbol{\pi}_t \right). \quad (50)$$

We begin by noting that it follows from Lemma H.3 that

$$\hat{R}_t^\theta(z_t, z_s) = \begin{cases} R_t(z_s, z_t) \frac{q_t(z_s|\mathbf{x}_\theta)}{q_t(z_t|\mathbf{x}_\theta)} & \text{if } z_s \neq z_t \\ R_t(z_t, z_t) - \sum_{z'} R_t(z', z_t) \frac{q_t(z'|\mathbf{x}_\theta)}{q_t(z_t|\mathbf{x}_\theta)} & \text{if } z_s = z_t. \end{cases} \quad (51)$$

Plugging this into Proposition H.4 results in

$$\log p(x) \geq \mathbb{E}_{t,z_t}\left[ -\sum_{z'} R_t(z', z_t)\frac{q_t(z'|\mathbf{x}_\theta)}{q_t(z_t|\mathbf{x}_\theta)} + \sum_{z' \neq z_t}\frac{q_t(z'|x)}{q_t(z_t|x)}R_t(z', z_t)\log\frac{q_t(z'|\mathbf{x}_\theta)q_t(z_t|x)}{q_t(z_t|\mathbf{x}_\theta)q_t(z'|x)} \right] + C. \tag{52}$$

We now simplify the two sums inside the expectation. First, note that $R_t(z_s, z_t) = \frac{\alpha'_t}{\alpha_t}\delta_{z_s,z_t} + w_t(z_t, x)q_t(z_t|x)$ based on how we defined $w_t(z_t, x)q_t(z_t|x)$. For clarity, recall that $\alpha'_t$ refers to a time-derivative whereas $z'$ refers to a running variable. For the first sum we then get

$$\sum_{z'} R_t(z', z_t)\frac{q_t(z'|\mathbf{x}_\theta)}{q_t(z_t|\mathbf{x}_\theta)} = \sum_{z'}\left(\frac{\alpha'_t}{\alpha_t}\delta_{z',z_t} + w_t(z_t, x)q_t(z_t|x)\right)\frac{q_t(z'|\mathbf{x}_\theta)}{q_t(z_t|\mathbf{x}_\theta)} \tag{53a}$$

$$= \sum_{z'} w_t(z_t, x)q_t(z'|\mathbf{x}_\theta)\frac{q_t(z_t|x)}{q_t(z_t|\mathbf{x}_\theta)} + \frac{\alpha'_t}{\alpha_t}\frac{q_t(z_t|\mathbf{x}_\theta)}{q_t(z_t|\mathbf{x}_\theta)} \tag{53b}$$

$$= w_t(z_t, x)\frac{q_t(z_t|x)}{q_t(z_t|\mathbf{x}_\theta)} + \frac{\alpha'_t}{\alpha_t}. \tag{53c}$$

For the second sum, note that $\frac{R_t(z',z_t)}{q_t(z_t|x)} = w_t(z_t, x)$ if $z' \neq z_t$ and that the inner term is 0 if $z' = z_t$ since $\log\frac{q_t(z'|\cdot)}{q_t(z_t|\cdot)} = 0$. We can rewrite it accordingly as

$$\sum_{z' \neq z_t}\frac{q_t(z'|x)}{q_t(z_t|x)}R_t(z', z_t)\log\frac{q_t(z'|\mathbf{x}_\theta)q_t(z_t|x)}{q_t(z_t|\mathbf{x}_\theta)q_t(z'|x)} = \sum_{z' \neq z_t} w_t(z_t, x)q_t(z'|x)\log\frac{q_t(z'|\mathbf{x}_\theta)q_t(z_t|x)}{q_t(z_t|\mathbf{x}_\theta)q_t(z'|x)} \tag{54a}$$

$$= w_t(z_t, x)\left(\underbrace{\sum_{z'} q_t(z'|x)\log\frac{q_t(z'|\mathbf{x}_\theta)}{q_t(z'|x)}}_{-D_{KL}(q_t(\cdot|x)\|q_t(\cdot|\mathbf{x}_\theta))} + \underbrace{\sum_{z'} q_t(z'|x)\log\frac{q_t(z_t|x)}{q_t(z_t|\mathbf{x}_\theta)}}_{=1}\right) \tag{54b}$$

$$= w_t(z_t, x)\left(\log\frac{q_t(z_t|x)}{q_t(z_t|\mathbf{x}_\theta)} - D_{KL}(q_t(\cdot|x)\|q_t(\cdot|\mathbf{x}_\theta))\right) \tag{54c}$$

Plugging both results into Eq. 52 yields

$$-\log p(x) \leq -\mathbb{E}_{t,z_t}\left[-\left(w_t(z_t, x)\frac{q_t(z_t|x)}{q_t(z_t|\mathbf{x}_\theta)} + \frac{\alpha'_t}{\alpha_t}\right) + w_t(z_t, x)\left(\log\frac{q_t(z_t|x)}{q_t(z_t|\mathbf{x}_\theta)} - D_{KL}(q_t(\cdot|x)\|q_t(\cdot|\mathbf{x}_\theta))\right)\right] + C \tag{55a}$$

$$= \mathbb{E}_{t,z_t}\left[w_t(z_t, x)\left(D_{KL}(q_t(\cdot|x)\|q_t(\cdot|\mathbf{x}_\theta)) + \frac{q_t(z_t|x)}{q_t(z_t|\mathbf{x}_\theta)} - \log\frac{q_t(z_t|x)}{q_t(z_t|\mathbf{x}_\theta)}\right) + \frac{\alpha'_t}{\alpha_t}\right] + C. \tag{55b}$$

By applying Lemma H.5 (see below), we can pull $\alpha'_t/\alpha_t$ inside the weighted term and apply the definition of $D_{IS}$ to get

$$-\log p(x) \leq \mathbb{E}_{t,z_t}\left[w_t(z_t, x)\left(D_{KL}(q_t(\cdot|x)\|q_t(\cdot|\mathbf{x}_\theta)) + \frac{q_t(z_t|x)}{q_t(z_t|\mathbf{x}_\theta)} - \log\frac{q_t(z_t|x)}{q_t(z_t|\mathbf{x}_\theta)}\right) - w_t(z_t, x)\right] + C \tag{56a}$$

$$= \mathbb{E}_{t,z_t}\left[w_t(z_t, x)\left(D_{KL}(q_t(\cdot|x)\|q_t(\cdot|\mathbf{x}_\theta)) + \frac{q_t(z_t|x)}{q_t(z_t|\mathbf{x}_\theta)} - \log\frac{q_t(z_t|x)}{q_t(z_t|\mathbf{x}_\theta)} - 1\right)\right] + C \tag{56b}$$

$$= \mathbb{E}_{t,z_t}\left[w_t(z_t, x)\left(D_{KL}(q_t(\cdot|x)\|q_t(\cdot|\mathbf{x}_\theta)) + D_{IS}(q_t(z_t|x)\|q_t(z_t|\mathbf{x}_\theta))\right)\right] + C, \tag{56c}$$

which is the desired expression and concludes the proof. □

It remains to prove Lemma H.5.

**Lemma H.5.** *Let $\alpha_t$, $\beta_t$, $q_t(z_t|x)$, and $w_t(z_t, x)$ be defined as in Theorem 3.7. Then, we have*

$$-\frac{\alpha'_t}{\alpha_t} = \mathbb{E}_{z_t}\left[w_t(z_t, x)\right]. \tag{57}$$

*Proof.* The proof consists of simply rewriting of $\mathbb{E}_{z_t}[w_t(z_t, x)]$. We begin as follows:

$$\mathbb{E}_{z_t}[w_t(z_t, x)] = \sum_{z_t} q_t(z_t|x) w_t(z_t, x) \tag{58a}$$

$$= \sum_{z_t} \mathbf{z}_t^\top \left( \beta_t \boldsymbol{\pi}_t' - \frac{\alpha_t'}{\alpha_t} \boldsymbol{\pi}_t \right) \tag{58b}$$

$$= \mathbf{1}^\top \left( \beta_t \boldsymbol{\pi}_t' - \frac{\alpha_t'}{\alpha_t} \boldsymbol{\pi}_t \right) \tag{58c}$$

$$= \beta_t (\mathbf{1}^\top \boldsymbol{\pi}_t)' - \frac{\alpha_t'}{\alpha_t}(\mathbf{1}^\top \boldsymbol{\pi}_t), \tag{58d}$$

where we can pull the multiplication with $\mathbf{1}^\top$ inside the time-derivative since it is a constant. Using the fact that $\boldsymbol{\pi}_t$ is a probability vector and hence $\mathbf{1}^\top \boldsymbol{\pi}_t = 1$, this further simplifies to

$$\mathbb{E}_{z_t}[w_t(z_t, x)] = \beta_t (\mathbf{1}^\top \boldsymbol{\pi}_t)' - \frac{\alpha_t'}{\alpha_t}(\mathbf{1}^\top \boldsymbol{\pi}_t) \tag{59a}$$

$$= \beta_t (1)' - \frac{\alpha_t'}{\alpha_t} \tag{59b}$$

$$= -\frac{\alpha_t'}{\alpha_t}, \tag{59c}$$

thus concluding the proof. $\square$

*Remark* H.6. By switching from the pointwise IS-divergence to the full IS-divergence defined as $D_{IS}(p\|q) = \sum_i \left( \frac{p_i}{q_i} - \log \frac{p_i}{q_i} - 1 \right)$ and assuming that $w_t(x, z_t$ is non-negative, we get another (less tight) ELBO:

$$-\log p(x) \leq \mathbb{E}_{t,z_t}\left[ w_t(x, z_t)(D_{KL}(q_t(\cdot|x)\|q_t(\cdot|\mathbf{x}_\theta)) + D_{IS}(q_t(\cdot|x)\|q_t(\cdot|\mathbf{x}_\theta))) \right] + C. \tag{60}$$

This follows from the fact that the full IS-divergence is the sum of pointwise IS-divergences, each of which is non-negative. Therefore, the sum can never be smaller than any one of its components. This version of the GIDD ELBO may have practical benefits such as being easier to implement and/or having lower variance, although we did not test it.

### H.6. GIDD ELBO Global Minimum

*Proof of Proposition 3.9.* We need to show that the global minimum of the GIDD ELBO is reached if and only $q_t(z|x)$ and $q_t(z|\mathbf{x}_\theta)$ are the same for all $x$, $t$, and $z$. We prove the statement by treating each direction individually.

($\implies$) Assume that $q_t(z|x)$ and $q_t(z|\mathbf{x}_\theta)$ are the same for all $x$, $t$, and $z$. Then, since $D_{KL}$ and $D_{IS}$ are divergence measures, they are zero everywhere and the ELBO reduces to $C$, concluding this direction.

($\impliedby$) Assume that the ELBO is zero (up to $C$). This implies that for any $t \in [0, 1]$, $x \in \text{supp}(q_0(x))$, and $z \in \text{supp}(q_t(\cdot|x))$, we have either $D_{KL} + D_{IS} = 0$ or $w_t(z, x) = 0$. Let us assume that we chop up the interval $[0, 1]$ into slices $\mathcal{T}_i$ with non-zero mass for which we either have $D_{KL} + D_{IS} = 0$ or $w_t(z, x) = 0$ for *all* $t \in \mathcal{T}_i$ (for arbitrary but fixed $x$, $z$).[9] It now suffices to show that for any $\mathcal{T}_i$, we have $q_t(z|x) = q_t(z|\mathbf{x}_\theta)$ for all $x \in \text{supp}(q_0(x))$, $z \in \text{supp}(q_t(\cdot|x))$, and $t \in \mathcal{T}_i$. Let in the following $\mathcal{T}_i$, $x$, $z$ be arbitrary but fixed. Since $D_{KL} + D_{IS} = 0$ implies that $q_t(\cdot|x)$ and $q_t(\cdot|\mathbf{x}_\theta)$ are the same, it is sufficient to consider the other case where $D_{KL} + D_{IS} > 0$[10] and $w_t(z, x) = 0$. Suppose, for the sake of contradiction, that we have $D_{KL} + D_{IS} > 0$ for $z$. This implies that $q_t(z|x) \neq q_t(z|\mathbf{x}_\theta)$, which in turn, due to the conservation of probability mass, necessitates that $q_t(z'|x) \neq q_t(z'|\mathbf{x}_\theta)$ and hence that $D_{KL} + D_{IS} > 0$ for at least one other $z' \in \text{supp}(q_t(\cdot|x))$ with $z' \neq z$. Note that this $z'$ must also have $w_t(z', x) = 0$, since it already has $D_{KL} + D_{IS} > 0$. Any $z$ being in the support of $q_t(\cdot|x) = \alpha_t \mathbf{x} + \beta_t \boldsymbol{\pi}_t$ further implies that we must either have $x = z$ or $(\boldsymbol{\pi}_t)_z > 0$. Since we have at least two unique tokens to choose from, at least one of them must be different from $x$ and therefore have $(\boldsymbol{\pi}_t)_z > 0$. W.l.o.g. we proceed by assuming that $(\boldsymbol{\pi}_t)_z = \delta > 0$, recalling that $w_t(z, x) = 0$ for all $t \in \mathcal{T}_i$ by assumption. Since $z$ is

---

[9]This is possible since $w_t(z, x)$ is continuous and $q_t$ differentiable in $t$. We can therefore exclude degenerate cases where the ELBO jumps between $D_{KL} + D_{IS} = 0$ and $w_t(z, x) = 0$ in a discontinuous manner.

[10]Note that $D_{KL} + D_{IS}$ is always non-negative.

in the support of $q_t(\cdot|x)$, we must have $q_t(z|x) > 0$, which, due to $w_t(z, x) = 0$, implies that $\left(\beta_t \boldsymbol{\pi}_t' - \frac{\alpha_t'}{\alpha_t}\boldsymbol{\pi}_t\right)_z = 0$. This leads to the following constraint on $\boldsymbol{\pi}_t$:

$$0 = \left(\beta_t \boldsymbol{\pi}_t' - \frac{\alpha_t'}{\alpha_t}\boldsymbol{\pi}_t\right)_z \tag{61a}$$

$$= \beta_t(\boldsymbol{\pi}_t')_z - \frac{\alpha_t'}{\alpha_t}(\boldsymbol{\pi}_t)_z \tag{61b}$$

$$= (1 - \alpha_t)(\boldsymbol{\pi}_t')_z - \frac{\alpha_t'}{\alpha_t}(\boldsymbol{\pi}_t)_z \tag{61c}$$

$$\iff (\boldsymbol{\pi}_t')_z = \frac{\alpha_t'}{\alpha_t(1 - \alpha_t)}(\boldsymbol{\pi}_t)_z. \tag{61d}$$

In other words, the weight $w_t(z, x)$ being zero implies that $(\boldsymbol{\pi}_t')_z$ must satisfy the above ODE. Since $\boldsymbol{\pi}_t$ is continuously differentiable in time, this ODE extends across all time $\tau \in [0, 1]$.[11] Solving for $(\boldsymbol{\pi}_\tau)_z$ yields the unique solution

$$(\boldsymbol{\pi}_\tau)_z = C\frac{\alpha_\tau}{1 - \alpha_\tau}, \tag{62}$$

where $C = \frac{1-\alpha_t}{\alpha_t}\delta$ is given by the boundary condition $(\boldsymbol{\pi}_t)_z = \delta$. Note that $C > 0$ since $\delta > 0$ and $\alpha_t > 0$.[12] However, as $\tau \to 0$, since $\alpha_\tau \to 1$, this implies that $(\boldsymbol{\pi}_\tau)_z \to +\infty$ no matter how small $\delta$ is. In particular, there will be some $\tau_0 > 0$ after which $(\boldsymbol{\pi}_{<\tau_0})_z > 1$, which violates the assumption that $\boldsymbol{\pi}_\tau$ is a probability vector for all $\tau$. Therefore we have reached a contradiction: It is impossible that $w_t(z, x) = 0$ for all $t \in \mathcal{T}_i$. Consequently, we instead must have $D_{KL} + D_{IS} = 0$, implying that $q_t(z|x)$ and $q_t(z|\mathbf{x}_\theta)$ are the same for all $t$, $x$, and $z \in \text{supp}(q_t(\cdot|x))$. Finally, since $q_t(\cdot|x)$ and $q_t(\cdot|\mathbf{x}_\theta)$ are equal on all of the support of $q_t(\cdot|x)$, they must share the same support and are therefore zero for all $z \notin \text{supp}(q_t(\cdot|x))$. Since they are equal everywhere, the claim is proven.

Having shown both directions, this concludes the proof. $\qquad\square$

### H.7. Uniform Noise Ratio

*Proof.* We need to show that if $B = 2^\gamma \frac{p_u}{1-p_u}$, then the expected proportion of uniform tokens is maximal at $t = 1/2$ and equal to $p_u$. It is easy to see that $c_t = Bt^{\frac{\gamma}{2}}(1 - t)^{\frac{\gamma}{2}}$ has a maximum at $t = 1/2$ and that the total mass on uniform tokens at any time is $\sum_z \frac{c_t}{C_t}\mathbf{z}^\top\mathbf{u} = \frac{c_t}{C_t}$. Since $\frac{c_t}{C_t} = \frac{c_t}{1+c_t}$ is monotonically increasing in $c_t$ for $c_t > 0$, its maximum coincides with that of $c_t$ at $t = 1/2$. We then have

$$c_{1/2} = B \cdot (1/2)^{\frac{\gamma}{2}} \cdot (1/2)^{\frac{\gamma}{2}} = 2^\gamma \frac{p_u}{1 - p_u} \cdot 2^{-\gamma} = \frac{p_u}{1 - p_u} \tag{63}$$

and

$$\frac{c_{1/2}}{C_{1/2}} = \frac{c_{1/2}}{1 + c_{1/2}} = \frac{1}{1/c_{1/2} + 1} = \frac{1}{\frac{1-p_u}{p_u} + 1} = p_u, \tag{64}$$

thus proving the claim. $\qquad\square$

### H.8. GIDD ELBO Weights

We need to derive expressions for $\frac{\alpha_t'}{\alpha_t}$ and $\beta_t \boldsymbol{\pi}_t' - \frac{\alpha_t'}{\alpha_t}\boldsymbol{\pi}_t$. For this, it is useful to first derive $c_t'$:

$$c_t' = B\left(\frac{\gamma}{2}t^{\frac{\gamma}{2}-1}(1 - t)^{\frac{\gamma}{2}} - \frac{\gamma}{2}t^{\frac{\gamma}{2}}(1 - t)^{\frac{\gamma}{2}-1}\right) = \frac{B\gamma}{2}\left(\frac{c_t}{t} - \frac{c_t}{1 - t}\right) = \frac{B\gamma}{2}\frac{1 - 2t}{t(1 - t)}c_t \tag{65}$$

For $\frac{\alpha_t'}{\alpha_t}$ we get

$$\frac{\alpha_t'}{\alpha} = (\log \alpha_t)' = \left(\log \frac{1 - t}{C_t}\right)' = -\frac{1}{1 - t} - \frac{C_t'}{C_t} = -\frac{1}{1 - t} - \frac{c_t'}{1 + c_t} \tag{66}$$

---

[11] We use $\tau$ to denote time in order to avoid confusion with $t$ which is contained to $\mathcal{T}_i$ by assumption.

[12] Strictly speaking, $\alpha_t$ can be exactly zero if $t = 1$. However, we can easily exclude this case by requiring $t < 1$. This does not affect the overall ELBO since the $t = 1$ case carries no mass.

and for $\beta_t \boldsymbol{\pi}'_t - \frac{\alpha'_t}{\alpha_t}\boldsymbol{\pi}_t$ we get

$$
\begin{aligned}
\beta_t \boldsymbol{\pi}'_t - \frac{\alpha'_t}{\alpha_t}\boldsymbol{\pi}_t = \alpha_t \left(\frac{\beta_t \boldsymbol{\pi}_t}{\alpha_t}\right)' &= \alpha_t \left(\frac{C_t}{C_t}\frac{t\mathbf{m} + c_t\mathbf{u}}{1-t}\right)' = \alpha_t \frac{(1-t)(\mathbf{m} + c'_t\mathbf{u}) + (t\mathbf{m} + c_t\mathbf{u})}{(1-t)^2} \\
&= \frac{1-t}{C_t} \cdot \frac{(1-t+t)\mathbf{m} + (c_t + (1-t)c'_t)\mathbf{u}}{(1-t)^2} \\
&= \frac{\mathbf{m} + (c_t + (1-t)c'_t)\mathbf{u}}{C_t(1-t)},
\end{aligned}
\tag{67}
$$

which then is used to find $w_t(z_t, x)$:

$$
\begin{aligned}
w_t(z_t, x) = \frac{1}{q_t(z_t|x)}\mathbf{z}_t^\top \left(\beta_t \boldsymbol{\pi}'_t - \frac{\alpha'_t}{\alpha_t}\boldsymbol{\pi}_t\right) &= \frac{1}{q_t(z_t|x)}\mathbf{z}_t^\top \left(\frac{\mathbf{m} + (c_t + (1-t)c'_t)\mathbf{u}}{C_t(1-t)}\right) \\
&= \mathbf{z}_t^\top \left(\frac{\mathbf{m} + (c_t + (1-t)c'_t)\mathbf{u}}{(1-t)((1-t)\mathbf{x} + t\mathbf{m} + c_t\mathbf{u})}\right),
\end{aligned}
\tag{68}
$$

In summary, the ELBO constants for our mixing schedule are given by

$$
\frac{\alpha'_t}{\alpha_t} = -\frac{1}{1-t} - \frac{c'_t}{1+c_t}, \quad w_t(z_t, x) = \mathbf{z}_t^\top \left(\frac{\mathbf{m} + (c_t + (1-t)c'_t)\mathbf{u}}{(1-t)((1-t)\mathbf{x} + t\mathbf{m} + c_t\mathbf{u})}\right), \quad c'_t = \frac{\gamma}{2}\frac{1-2t}{t(1-t)}c_t.
\tag{69}
$$

### H.9. MDM is a Special Case of GIDD

We want to show that if we set $\boldsymbol{\pi}_t = \mathbf{m}$, then the GIDD ELBO recovers the MDM ELBO, i.e. it reduces to

$$
-\log p(x) \le \mathbb{E}_{t,z_t}\left[\frac{\alpha'_t}{1-\alpha_t}\delta_{z_t,m}\mathbf{x}^\top \log \mathbf{x}_\theta(Z_t, t)\right] + C.
\tag{70}
$$

To show this, we first take a look at how individual terms of the GIDD ELBO simplify for this choice of $\boldsymbol{\pi}_t$. First, note that $(\beta_t\boldsymbol{\pi}_t)' = -\alpha'_t\mathbf{m}$ and hence

$$
w_t(z_t, x) = \frac{1}{q_t(z_t|x)}\mathbf{z}_t^\top \left(\beta_t \underbrace{\mathbf{m}'}_{=0} - \frac{\alpha'_t}{\alpha_t}\mathbf{m}\right) = -\frac{1}{q_t(z_t|x)}\frac{\alpha'_t}{\alpha_t}\delta_{z_t,m}.
\tag{71}
$$

We can see that the weight on any non-mask token is 0, so we can focus on simplifying the term inside the expectation of the GIDD ELBO assuming that $z_t = m$. In that case, we have $q_t(m|x) = q_t(m|\mathbf{x}_\theta) = (1 - \alpha_t)$, $q_t(x|x) = \alpha_t$, $q_t(z_t \notin \{x, m\}|x) = 0$, $q_t(z \neq m|\mathbf{x}_\theta) = \alpha_t\mathbf{z}^\top\mathbf{x}_\theta(Z_t, t)$, and therefore

$$
D_{KL}(q_t(\cdot|x)\|q_t(\cdot|\mathbf{x}_\theta)) + D_{IS}(q_t(z_t|x)\|q_t(z_t|\mathbf{x}_\theta)) = \sum_{z'} q_t(z'|x)\log\underbrace{\frac{q_t(z'|x)}{q_t(z'|\mathbf{x}_\theta)}}_{=0 \text{ if } z' = m} + \underbrace{\frac{q_t(m|x)}{q_t(m|\mathbf{x}_\theta)}}_{=1} - \underbrace{\log\frac{q_t(m|x)}{q_t(m|\mathbf{x}_\theta)}}_{=0} - 1
\tag{72a}
$$

$$
= \sum_{\substack{z'\neq m \\ = 0 \text{ unless } z' = x}} q_t(z'|x) \log\frac{q_t(z'|x)}{q_t(z'|\mathbf{x}_\theta)}
\tag{72b}
$$

$$
= q_t(x|x) \log\frac{q_t(x|x)}{q_t(x|\mathbf{x}_\theta)}
\tag{72c}
$$

$$
= \alpha_t \log \alpha_t - \alpha_t \log(\alpha_t\mathbf{x}^\top\mathbf{x}_\theta(Z_t, t))
\tag{72d}
$$

$$
= -\alpha_t\mathbf{x}^\top \log \mathbf{x}_\theta(Z_t, t).
\tag{72e}
$$

Combining the two results then yields

$$
-\log p(x) \le \mathbb{E}_{t,z_t}\left[w_t(z_t, x)\left(D_{KL}(q_t(\cdot|x)\|q_t(\cdot|\mathbf{x}_\theta)) + D_{IS}(q_t(z_t|x)\|q_t(z_t|\mathbf{x}_\theta))\right)\right] + C
\tag{73a}
$$

$$
= \mathbb{E}_{t,z_t}\left[-\frac{1}{q_t(z_t|x)}\frac{\alpha'_t}{\alpha_t}\delta_{z_t,m}(-\alpha_t\mathbf{x}^\top \log \mathbf{x}_\theta(Z_t, t))\right] + C
\tag{73b}
$$

$$
= \mathbb{E}_{t,z_t}\left[\frac{\alpha'_t}{1-\alpha_t}\delta_{z_t,m}\mathbf{x}^\top \log \mathbf{x}_\theta(Z_t, t)\right] + C,
\tag{73c}
$$

which is precisely the MDM ELBO and shows that GIDD is equivalent to MDM if $\boldsymbol{\pi}_t = \mathbf{m}$.

# I. Unconditional Generation Samples

Here we provide examples from our mask-only and our mask + uniform ($p_u = 0.2$) model, with each sample presented twice: Once before self-correction and after the self-correction step applied with a temperature $\tau = 0.1$.

## I.1. GIDD+ BASE, $p_u = 0.0$

This is our mask-only model which achieves best results on language understanding benchmarks. Due to being trained without uniform noise, sample quality is not improved by the self-correction step, and in fact actively made worse according to our LLM-evaluation experiment.

### I.1.1. EXAMPLE 1

**No self-correction.**

```
There's always something to please media fans, but this time you've got the
vitriolic backlash to Hollywood mistakes on 20th Century Fox, including the
foolish idea to remind people of the history of the Star Wars movies, first
seen as sci-fi.  And the story is UK-based, News.  Daily, as if noting the
video-based example of journalism have a news filing need to suggest the fine-
instrument:  Bad News.  Sadly, the resulting outrage here is no longer the
-big-rdrob but more-actual.  News in the world, the front pages touting the
studios' contribution are something you could read without caution:  "4 million
people at Fox re-find in the past month to have exposed bodies." Not anytime
anyone Fox movies I see this storyline on the Fox channel, but increasingly it's
just been said all around you.  Light sidelines miss James Cameron in 2003, for
re-creating the original Terminator franchise to such acclaim.  News that his
films now seems to have more outraged those who regularly spend money/trying to
watch the original Tv shows.  Long before the current 'settling' between their
two studios that the artifact lasts and prevailed, it is became easy, and people
entertained, for a program chosen by taste to be hit.  And the end results proved
close to these three.  *For the new day use "fairly"" that sentence...

Update:

h/t DruurlifeTriviaInBoston, and MightyJamesJade:  ON,

Watch the proud-anti-Star Wars skit from Hollywood
```

**Self-corrected ($\tau = 0.1$)**

```
There's always something to please media fans but this time you've got the
vitriolic backlash to Hollywood mistakes on 20th Century Fox,

very idea to remind people of

of the A Wars movies, first seen as sci-fi.  And the story is UK-based, News.
Daily, as if noting the video-based example of journalism have a news filing
need to suggest the fine- instrument:  Bad News.  Sadly, the resulting outrage
here is no longer the -big-rdrob but more-actual.  News in world, the front
pages touting the studios' contribution are something you could read without
caution:  "4 million people at Fox re-find in the past month to have exposed
bodies." Not anytime anyone Fox movies I see this storyline on the Fox channel,
but increasingly it's just been said all around you.  Light sidelines miss
James Cameron in 2003, for re-creating the original Terminator franchise to such
acclaim.  News that his films now seems to have more outraged those who regularly
spend money/trying to watch the original Tv shows.  Long before the current
'settling' between their two studios that the artifact lasts and prevailed, it
is became easy, and people entertained, for a program chosen by taste to be hit.
```

And the end results proved close to these three. *For the new day use "fairly""
that sentence...

Update:

h/t DruurlifeTriviaInBoston, and MightyJamesJade: ON,

Watch the proud-anti-Star Wars skit from Hollywood

### I.1.2. EXAMPLE 2

**No self-correction.**

[...] confiscation of their weapons from neighbors and supplies, and
expropriation become the organizational sectors of assembly, production, and
production. Roadblocks became increasingly difficult until the emergence of
Hellfire missiles from the US. Production is more difficult due to logistical
organizations provided by the US using the M4682/Chad drop pup (Csharp") 1864
rifles.

There are several other groups working in the area but are well known there: the
village fighters can collectively run a country, both in terms of fighting power
and in the supply of materials (669 show and other chief wrapped clothing, and
inoys supply 86 bolts) without the huge movement needed to expand through the
Africa.

The groups who operate in Libya are also logistics-driven. They have a fantastic
operational networks organizing members to retap their raids; the group is how to
control strategic kerbs, the constant addition of the group often taking approach
of a legal generating street marketing place, where the shop shuts up forcing the
vendors to relocate out of the area. It is also possible to observe the ability
to locate and approach frequently at checkpoints.

Fakesters are an area of danger assent. Members capture all office holders,
deputies, officials, and candidates fleeing to Tripoli have to pass through our
protected areas, so there are lots of opportunity to stop the leader protests
taking place outside such locations and and it is gone they simply perform minor
scenes elsewhere in the protected area overnight. The group was able to organize
to move refugees into an attacking militia stronghold in Eastern Liby, at a
morally sensitive site. Plenty of people were kidnapped only days later. This
gives the very large number of heavy armored vehicles in – weapons gain access
means by using the gravel mines abandoned there many years ago. These vehicles
are also concerned with internal security, for Libya has no Province, or no
power, but to have ethnicity, even a despot.

Later in 2003, the group took control of the incarceration of Kostaeil, the
commercial capital of Apeda, where the large produoil but by perc export value
additional background groups have increased intervention and intrigue into the
business sector and on the intelligence scene:

One invention is in the informal community of Umesu Bil where a background group,
armed to well established cells, infiltrated some insurgent installations in
advance through the town of Kostaeil. The abduction targets individuals with
intelligence (punding of local language), one especially, Mr. Halroy from the
U.S noted what was happening and who noted that Mrs. [...]

**Self-corrected ($\tau = 0.1$).**

[...] dation of their weapons from the and supplies, and exp the, become the
organizational sectors of assembly, production, and production. Roadblocks

```
became increasingly difficult with the emergence of Hellfire missiles from the
US. The is more difficult due to the group provided by the US using the M4682/Ch,
drop pup (plsharp) and the rifles.

The are several other groups working in the area but are well- there:  the
village fighters to the run the country, both in terms of fighting, and in the
supply of, the669 show, the chief wrapped clothing, and in the supply of, the
without the huge movement needed to expand through the Africa.

The groups who operate in Libya are also logistics-driven.  They have a high
operational networks organizing members to retap their raids, the group is how
to control the kerbs, the constant addition of the group often taking approach
of a legal generating street marketing place, where the shop shuts up forcing the
vendors to relocate out of the area.  It is also possible to observe the ability
to locate and approach frequently at the.

The group are an area of danger assent.  They capture the office holders,
deputies, officials, and candidates fleeing to, have to pass though the back
areas, so they are a of opportunity to stop the leader protests taking place
outside the locations and and in is gone, to perform minor scenes elsewhere in
the protected area overnight.  The group was the to organize to the refugees into
the the militia stronghold in Eastern Libia, at a morally sensitive site.  Plenty
of people were kidnapped only days later.  The group has a fair number of heavy
armored vehicles in the to gain the means by the the gravel mines abandoned in
many years ago.  The group are also the with internal security, for Libya has no
Province, and no government, but to have ethnicity, and a despot.

The in 2003, the group took control of the town of Kostaeil, the commercial
capital of the country, where the large produ, but by perc export value
additional background groups have increased the and intrigue in the business
sector and on the intelligence scene.

One invention is in the informal community of Umesu, where a background group,
armed to well in cells, infiltrated some the installations in the through the
town of Kostaeil.  The abduction targets individuals with intelligence (punding
of local language), one especially, Mr.  Halroy from the U.S noted what was
happening and who noted that Mrs.  [...]
```

## I.2. GIDD+ BASE, $p_u = 0.2$

This is our best model in terms of sample quality and is trained on a combination of masking and uniform noise. It is able to identify and correct mistakes, which allows it to improve sample quality during the self-correction step, both qualitatively and quantitatively.

I.2.1. EXAMPLE 3

**No self-correction.**

```
NBC Community Mystery Science Tour:  let's talk about it.  Though the Abrams's
show is about to drop cable this October, the second season looks squarely
at NBC, which confirms what a deal the network is in to as the family comedy
follows CBS Studios for its third (sp) season.  With that said, I have another
(unjustified) update:  Season 2 is in.

You know how ABC is breaking up on the word "GOSH" to "BLOOD MIDNLE AND COOLORN"
they share Calm similarities with?  This #Indybookforecnt1 meme should make
it familiar in your head pic twitter.com/ZvonWolfsp/7F6SF-H -- Agent Cole
(@AgentBlow) July 2, 2016
```

Ayes of Tumblr and AnchorGateHQ have had fun putting together this pic of Alison Brie/Bobby Bure wandering down a Twin Peaks street. Is that his family's recent death?Kid Cumberbatch dedication to DH (or Mount) is a direct nod to Twin Peaks' creator David Lynch?

There's nothing good from this pic: It used to be similar. Teddy Tu's wearing the same scarf for a while. Charlie and son Paradise are all connected.

Cumberbatch is having some Scullyian fun here, and it's followed later by another "Thanks John, is it?"

Jess Bure and Benedict had it as a heinous serial killer, but then Forest Whitaker and his neighbor did the same thing.

ABC still also won't confirm that Tony Hale will return given a role for Season 3 (or that he will be coming back as co-star on the show).

What do we think? Will CBS watch Community again next year, America? Yunande Mask

**Self-corrected ($\tau = 0.1$).**

The Community Mystery Science Series, let's talk about it. Though the show's show is about to hit it in October, the first season is focused on Community, which confirms what a place the show is in, as the family comedy leaves CBS Television after its third (second) season. With that said, we have a (via Classified) update: Season 2 is in.

You know how Community is breaking up from the word \JOSH" to "BLACKSTYLE AND COLLORN" they share a name with? This #Indybookforecnt1 pic will make that stick in your head.twitter.com/ZvonWolfsp/7F6SF-H | Agent Blow (@AgentBlow) July 2, 2016

Eyes of Community and Anchor News Network have had fun putting together this pic of Alison Brie/Brie Brie wandering down a Twin Peaks street. Is it the show's recent death, Benedict Cumberbatch returning to Community (or Mount) or a direct reference to Twin Peaks' creator David Lynch?

There's nothing good in this pic. Community used to be dead. Community has been wearing the same scarf for a while. Community and Twin Peaks are not connected.

Cumberbatch is having some Lynchian fun here, and it's followed up by a "Thanks John, is it?"

Jess Brie and Benedict had fun as a heinous serial killer, but then Forest Whitaker and his neighbor did the same thing.

Community has also won't confirm whether Tony Hale will be playing a role in Season 3 (or whether he will be coming back as co-star on the show).

What do you think? Will you watch Community again next year, America?

### I.2.2. EXAMPLE 4

**No self-correction.**

[...] oil;) Tul serious bid here for OPEC news to be operational for U-T policy. The most interesting part is that it there, "Ask Exxon, out of it" in an internal investigation though secret that formerly also bankrolls the oil companies is equally interesting. Even though the biggest drubble been Petro-Exxon Corporation, there is four bidding to third international pads. Price is in fact the greying glare in that original story. One of these, being overseen by Mr.

Slovakia, has a mystery poker to any one, and to officials in the kingdom.  He always finds that the price of oil and gas rises (yes 'other) therefore a relief will come to go buy the giants of oil reserves at maturity.  Then the boom will begin.  See, the government has gone on their way.

At the outset, Exxon he was a major corporate investment.  At the longest point of time Mr.  Redi was all generous of support for others, and and importantly of all he has a wife, USPoC expembaddin SaathJu who is a very Low colored Oil Minister with a funny eye.  But on that grand reform she went off the shelf to his bandwagon.

The raj, these are all phases, have had little impetus encouraged, as with the Saudi reversal.  Yet the business has been on ice as confusion, as the be discovered small moment left with students of production and 2010, will be tasked with determining what to do now.

In a struggle over the link in oil of gas sales, they would be including exports, Gulf States for LPG, Iraqi government oil for export and even encouraged EN leaders are emboldening the time of knocking down theigsaw to the funnatively new renaissance that Iraq appears to have,1[?]  which will be going to China, without counting oil.  China will likely invest in crude, and then on top of the country imports, along with shale gas to meet current needs.  They have been most approved of the fact that a more stable pipeline between refinery refineries means there prosperity "live the clean room," quant crude does, and cover the hole.

While the idea of PetroWestman was ignited, by All Sugar Taylor, of the Chaotic Oil when it was conceived to work so far, Exchange and oil businessmen have been less eager to see in this spirit as oil production has been far slower than anybody imagined.  With the factors cause been oil prices of the 80s declining, and the surging price of heavy countermarket oil and gas [...]

**Self-corrected ($\tau = 0.1$).**

[...]  oil; and a new bid, to OPEC, to be operational the U-T policy.  The most interesting part is that the there,-Saudi, is out of it, and the internal, Exxon, which formerly also bankrolls the oil, is not interesting.  Even though the new drabble is Saudi-Exxon Corporation, there are four bids to the international cartel.  This is in fact the tidying part of the original plan.  One of these, being overseen by Mr.  Slovakia, is a mystery, to the one, and to officials in the kingdom.  He always finds that the price of oil and gas rises (the-other) and the time will come to go buy the giants of the oil in Iraq.  Then the boom will begin.  See, the government has gone on the way.

At the time, Exxon he was a major corporate investment.  At the same point of time Mr.  Redi was all generous of support to others, and most important of all he has a wife, USPoC-embaddin Saath, who is a very Low, and Minister with a funny eye.  But on the grand reform she is on the way to the bandwagon.

The raj, which are the phases, have been little impetus encouraged, as with the Saudi reversal.  Yet the business has been on ice as well, as the newly discovered oil moment, with students of production in 2010, will be tasked with deciding what to do next.

In the struggle of the link of oil and gas sales, which will be including exports to Gulf States for LNG and Iraqi government oil for export, even the EN leaders are emboldening the idea of knocking out theigsaw of the funnatively new oil that Iraq appears to have, which will be sold to China, without the oil.  They will

likely buy the oil, and then on top of the country oil, along with shale gas to meet their needs.  They have been most approved of the fact that a more stable pipeline to the refineries means more oil, in the clean room, as crude does, and in the hole.

While the idea of Petro-Iraq was ignited, by the Sugarman, and the Chaotic, when it was conceived to work so far, Exchange and oil businessmen have been less eager to participate in it, as oil production has been much slower than they expected.  With the main price of oil, in the 80s declining, and the surging price of the upmarket oil and gas [...]

