# OpenReview forum: "Generalized Interpolating Discrete Diffusion"
_ICML.cc/2025/Conference — ICML 2025 poster_

### Official Review · Reviewer_LeK6 · 2025-03-10

**Overall Recommendation:** 3

**Summary:**

The method overcomes existing limitations in autoregressive models and discrete diffusion approaches by introducing a generalized interpolating discrete diffusion. This innovation offers enhanced flexibility in the noising process design by combining masking and uniform noise, enabling the revision of previously generated tokens.

### After rebuttal comment
Overal, the paper is of sufficient quality to warrant acceptance given how flexible the method enables for self-correction, which is also an important aspect of not just diffusion methods but also AR models. That's said, the concerns regarding overfitting problem is not fully addressed and I believe the authors should be upfront about this issue. Since looking at table 4, the model with less training iterations obtain better metrics than the longer one, yielding a concern regarding the scalability of the model on large-scale training like bigger model size, large-scale dataset. In addition, the evaluation of self-correction using LLM-based metrics should be included in the final manuscript and discussed thoroughly. Given these factors, I decide to keep my current rating.

**Claims And Evidence:**

The problem of keeping already generated tokens unchanged is clearly presented, highlighting the essence of the proposed method. However, to revise generated tokens, the method still relies on a uniform diffusion process that is not deterministic and controllable. Regarding the self-correction step, the resampling strategy for low-confident tokens is similar to the MaskGIT paper.

**Essential References Not Discussed:**

NA

**Experimental Designs Or Analyses:**

- Should not let empty cells in table 4.

**Methods And Evaluation Criteria:**

See above.

**Other Comments Or Suggestions:**

NA

**Other Strengths And Weaknesses:**

Strength: Paper is well-written and easy to follow.

Weakness: See questions below.

**Questions For Authors:**

1. The use of uniform noise for mask diffusion is also introduced as a mask-and-replace transition in "Vector Quantized Diffusion Model for Text-to-Image Synthesis" paper. A discussion should be included.
2. The proof of proposition 3.3. should be put in the supplementary as it is not a significant part of the paper.  And It is quite similar to the derivation of existing works like "Simplified and generalized masked diffusion for discrete data", "Structured denoising diffusion models in discrete state-spaces". Section 3 appear to present background information rather than the paper's technical contribution, excluding the ELBO 3.3.
3. Self-correction step is similar to the sampling strategy in MaskGIT paper (MaskGIT: Masked Generative Image Transformer).
4. What is the method's performance when ignoring weighting term? This mean the weight is set to a constant 1.
5. Table 4 raises questions about GIDD's performance when trained for longer periods, as the results suggest potential overfitting. It's unclear why the authors include other baselines with 1.1B parameters while omitting results for their method at the same parameter size, making direct comparisons difficult.
6. How does large uniform noise (e.g., 0.5) influence model performance?
7. Table 3 demonstrates that the use of uniform noise does not improve perplexity (PPL). An alternative approach would be to report entropy instead.
8. Since the method has an additional self-correction step, I wonder it will introduce a latency for the sampling process? A report of sampling speed is needed (e.g. tokens/sec)

**Relation To Broader Scientific Literature:**

NA

**Theoretical Claims:**

NA

---

> ### Author Rebuttal · Authors · 2025-03-29
>
> We thank the reviewer for their feedback and insightful questions. We especially feel that 2) deserves careful discussion since we consider the theoretical contributions a key part of our work. We would like to encourage the reviewer to share some additional detail on their concerns.
>
> 1. We agree with the reviewer that a discussion of [4] needs to be added. While [4], and also [2] (App. A.2.6), experiment with a BERT-like combination of masking + uniform noise, these approaches are only discrete-time and do not investigate what improvements/challenges arise from it.
> 2. We agree that there are some similarities between our theoretical contributions and the suggested references [1, 2] and would like to highlight the important differences. The theory behind GIDD is a strict generalization of masked diffusion (MD) in [1] (see Cor. 3.8). Unlike MD, GIDD allows for a time-variable mixing distribution $\pi_t$, which makes MD a special case where $\pi_t=\mathbf{m}$ is constant w.r.t. time. Another example would be setting $\pi_t=\mathbf{1}/|V|$ where $|V|$ is the vocabulary size, which simplifies to uniform diffusion from [2]. As for [2], this paper covers the most general case of discrete diffusion and forms the foundation of much subsequent work, including [1] and ours. In this framework (and in discrete diffusion in general), complete knowledge of the transition matrix is paramount and necessary both for computing the ELBO as well as for sampling. In this regard, our theoretical contribution is to solve the inverse problem of finding the (closed-form) Markovian transitions given only the marginal forward transitions for a special family of linearly interpolating diffusion processes (see Prop. 3.3). In addition, while [2] only covers discrete-time, the GIDD ELBO and Markov transitions are stated in continuous-time by applying the tools from [3]. We hope that this sufficiently highlights our theoretical contributions and that we were able to address the concerns of the reviewer.
> 3. We agree with the reviewer that the confidence-weighting in the self-correction step is similar to the MaskGIT sampling algorithm. However, ours is a fixed-point iteration that resamples one token at a time and is only applied after denoising, whereas MaskGIT sampling is an adaptive masked diffusion sampler prioritizing high-confidence tokens. Indeed, applying MaskGIT sampling to GIDD ($p_u=0.0$; base) finds that this is prone to collapse, with generated samples being low-diversity and consisting mostly of repeated tokens. This manifests in a low generative entropy (via Gemma-2-9B) of 1.31 compared to the baseline of 3.13. MaskGIT sampling is unfortunately not applicable to $p_u>0$ models since those require token replacements in addition to unmasking. We thank the reviewer for pointing out this similarity and will discuss it in an updated version.
> 4. Preliminary experiments showed that setting the ELBO weights to 1 works for $p_u=0.0$ but is challenging for $p_u>0$, and in light of the clipping strategy working much better, we did not investigate this any further. We hypothesize that the weight ratio between uniform and noise-free tokens is important for learning the correct posterior: What is the distribution of the correct token, given that it is incorrect with only a small probability?
> 5. Regarding overfitting, we would like to point out that for $p_u=0.0$, MDM and GIDD are equivalent, so it is likely that if GIDD is overfitting, then the baselines are also overfitting. One potential cause could be the way long sequences are handled in our data loader (random cropping instead of splitting), which leads to more frequent repetition of short sequences, potentially leading to overfitting. The 1.1B MDM baseline is included purely for context as training 1.1B GIDD models was unfortunately not feasible given the available resources.
> 6. Discouraged by the higher loss and PPL of $p_u>0$ models, we did not experiment much with noise levels above $p_u=0.2$. Generally speaking, higher values are more challenging as the SNR drop gets more concentrated on early steps.
> 7. Unfortunately, we are not entirely sure which entropy the reviewer is referring to. Since PPL is just the exponent of cross-entropy, the comparison would not qualitatively change by reporting cross-entropy.
> 8. Since self-correction is run for at most 128 iterations (subject to early stopping) and the samples are generated in 128 denoising steps, the worst-case computational overhead is double. However, even if self-correction incurs some overhead, it is generally possible to improve sample quality while keeping the inference compute budget constant by reducing the number of denoising steps in favor of additional post-hoc correction steps.
>
> - [1] Shi et al., 2024. https://arxiv.org/abs/2406.04329
> - [2] Austin et al., 2021. https://arxiv.org/abs/2107.03006
> - [3] Campbell et al., 2022. https://arxiv.org/abs/2205.14987
> - [4] Gu et al., 2021. https://arxiv.org/abs/2111.14822

---

> > ### Comment · Reviewer_LeK6 · 2025-04-04
> >
> > Thank the authors for the response! Here are my follow-up questions:
> > 1. Regarding Q7, I mean the entropy of the model's generated samples. Though the paper proposes mixing noise (including mask and uniform noise) that supports the ability of token revision, it lacks strong evidence to support the claim. So, additional results are (both quantitative and qualitative) encouraged. This stems from the fact that PPL does not reflect the "true" quality of generated samples.
> > 2. Could you provide that actual inference speed (i.e. token/sec) in comparison with MDM baseline? It is essential to include this aspect for the sake of paper completeness.
> > 3. Regarding the absence of the 1.1B model: I believe that whatever is included in a paper does count and has it purpose. If the 1.1B model is not available at the submission, at least the authors should mention it somewhere or just discard it. Still, it is nice to have.
> > 4. The explanation of overfitting is not quite convincing to me. Is it just solely about data processing or model itself? If it is due to the model, the authors should admit it in the limitation section.

---

> > > ### Author Response · Authors · 2025-04-06
> > >
> > > We thank the reviewer for their rebuttal comment and will address their follow-up questions in the following.
> > > 1. To prevent any potential confusion, we would like to clarify that there are two different PPLs used throughout the paper: Validation PPL (model's PPL on the validation set) is the upstream metric, whereas generative PPL (PPL of Gemma-2-9B on generated samples) is the downstream metric. It is common to ablate training settings on the upstream metric (i.e. val. PPL), which is what is reported in Tables 2 and 3. The entropy of generated samples is also a downstream metric, so using it for ablations would be unusual and somewhat impractical. However, it is well-suited for evaluating the final GIDD+ checkpoints and for comparing different levels of $p_u$, which we have done in our reply to Reviewer wojY (Q2, Table 1). Regarding the request for additional results, we would like to point towards Q3 and Table 2 in our response to Reviewer wojY, where we have conducted an LLM-based evaluation of the self-correction abilities of the different noise schedules.
> > > 2. When generating 128 samples with a batch size of 1 and 128 denoising steps, we get a sampling speed of 1.27 sec/seq for GIDD and 1.18 sec/seq for MDM on a RTX 4090. This translates to a speed of 404 tok/sec and 433 tok/sec respectively. The slight overhead (~7%) of GIDD stems from handling general-case mixing distributions and is constant w.r.t. the model, so it will shrink as the model size increases. Since the architecture is shared between GIDD and MDM, model inference takes the exact same amount of time. It is also important to mention that even in the absence of any self-correction steps, the gen. PPL of $p_u>0$ models is significantly better than $p_u=0.0$ and MDM (see Fig. 6, App. D), making it worth the additional cost. Given the slight overhead imposed by GIDD, we agree with the reviewer that this is important to mention and will include it in future revisions of the paper.
> > > 3. While we are working on scaling up the proposed models, we cannot promise any results on that front in the near future, given that the primary constraint was and is the availability of computational resources. In any case, we will make sure to mention this in future revisions.
> > > 4. It is important to highlight that overfitting results from our data processing and is not model specific--if it is happening at all. As Reviewer wojY correctly points out, the differences may be too small to even make such a claim.

---

### Official Review · Reviewer_wojY · 2025-03-11

**Overall Recommendation:** 3

**Summary:**

As a class of models currently attracting significant attention, masked diffusion models suffer from a fundamental limitation: once a token is generated, it cannot be modified. To address this issue, this paper introduces General Interpolating Discrete Diffusion (GIDD), which allows for a more flexible noise formulation. The authors derive the forward process, backward process, and loss function for GIDD. Additionally, they propose several techniques to stabilize training and improve performance. Experimental results show that while GIDD increases the modeling difficulty—leading to a performance drop in language modeling perplexity and downstream tasks—it effectively demonstrates self-correction capabilities.

**Claims And Evidence:**

The claims made in this paper are not well supported by the experimental results.

Specifically, language modeling perplexity and downstream task performance are the primary metrics of interest, yet the proposed method leads to worse performance on both.

Regarding generative PPL, this metric is not entirely reliable. Even low-quality sentences (e.g., repetitive words) can receive low PPL from large language models, a limitation that the authors themselves acknowledge. Given this, the advantage demonstrated by GIDD is only observed on this unreliable metric, raising concerns about the validity of the claimed improvements.

**Essential References Not Discussed:**

No.

**Experimental Designs Or Analyses:**

Yes. Please see Claims And Evidence.

**Methods And Evaluation Criteria:**

No. As discussed in Claims and Evidence, the primary evaluation metrics (language modeling PPL and downstream task performance) show a decline, while the generative PPL metric used to support the method is unreliable. Thus, the evaluation does not convincingly justify the claims.

**Other Comments Or Suggestions:**

1. In Section 5.3, the authors claim that GIDD trained on only 131B tokens surpasses models trained for twice as long, attributing this to overfitting on spurious patterns in the training data. However, this explanation seems questionable, as such results are more likely due to random variation rather than overfitting. Moreover, on datasets such as ARC-c, BoolQ, OBQA, and WinoG, both models perform close to random guessing, making it difficult to determine which is superior.

2. When using generative perplexity as an evaluation metric, I suggest including entropy as a complementary measure to assess the diversity of generated sentences, providing a more comprehensive perspective.

3. To better evaluate the quality of generated text, I recommend conducting a user study or leveraging LLM-based scoring, as these approaches would offer more reliable and interpretable assessments.

4. Would it be possible to include results on reasoning benchmarks, such as GSM8K?

If the author provides the aforementioned more detailed evaluation metrics (or at least some of them), I will increase my score.

**Other Strengths And Weaknesses:**

The paper has a clear motivation, well-structured theoretical derivations, and is written in a clear and concise manner.

However, the main weakness is that the experimental results do not sufficiently support the effectiveness of the proposed method (as detailed in Claims and Evidence). Although the authors introduce "self-accuracy" as an evaluation metric, they do not provide sufficient justification for its validity.

**Questions For Authors:**

1. In Figure 3, what does "Tokens changed" represent? How does it relate to the number of sampling steps?

2. For GIDD with uniform noise, is it possible to set the number of sampling steps to an arbitrarily large value? I am particularly curious whether tokens continue to change in the later stages of sampling when the number of steps is very large.

**Relation To Broader Scientific Literature:**

Discrete diffusion models have recently made significant progress in text generation, particularly masked diffusion models. However, a key limitation of masked diffusion models is that once a token is generated, it cannot be modified—an inherent constraint of their noise injection process. Addressing this issue has been a major focus in the field.

This paper introduces a generalized forward process that combines masked noise and uniform noise, representing a valuable exploration of this problem.

**Theoretical Claims:**

No. I did not verify every step of the derivations, but the theoretical arguments appear logically sound and well-structured.

---

> ### Author Rebuttal · Authors · 2025-03-31
>
> We thank the reviewer for their insightful review and constructive feedback. In the following, we would like to respond to the reviewer’s comments and suggestions and provide additional experimental results to further bolster the claim of self-correction in the presented models.
>
> 1. We mostly agree with the reviewer on the comment regarding overfitting. The difference is indeed small, and performance is almost random on some datasets. Given that the difference is consistent, albeit small, perhaps the appropriate characterization would be to say that it is likely run-to-run variance with some likelihood of overfitting. As mentioned in our response to Reviewer LeK6, we see a potential cause of overfitting in the way long sequences are handled in our data loader, which leads to more frequent repetition of short sequences, potentially causing overfitting.
> 2. We agree with the reviewer that this would be a good addition and would help quantify the diversity loss during self-correction. Given the mode-seeking nature of the self-correction step, some decrease in diversity is to be expected, but this should be restricted to a reasonable amount. Indeed, we find a decrease in entropy for $p_u>0$ models, with the decrease correlating linearly with the number of changed tokens (see Table 1 below). However, the decrease is moderate and does not indicate any collapse. This is also supported by the LLM-based evaluation (see 3.). For further context, we also provide qualitative self-correction examples in Appendix K, which give some intuition on the nature and extent of the reduced diversity.
> 3. Unfortunately, conducting a user study comes with its own host of challenges and is beyond the scope of this project. However, we have conducted an LLM-based evaluation of the quality of generated samples via GPT-4o using single-answer grading [1] (prompt omitted due to char. limit). The samples are graded on a scale from 1 to 10 in terms of clarity, grammaticality, factuality, writing style, and creativity. We would like to emphasize that these absolute numbers are highly dependent on the judge model’s calibration and should therefore be taken with a grain of salt. Nevertheless, we find consistent improvements at high significance levels, with $p_u=0.2$ exhibiting both the largest improvement and the highest scores overall (see Table 2 below). We report the self-correction setting with the largest effect for each noise level.
> 4. Given that our models are comparatively small and not trained on instruction following or conditional generation, evaluation on GSM8k is unfortunately not possible. Even if we try to do pseudo-conditional generation by forcing the logits of prompt tokens, the model does not generate meaningful continuations.
>
> Regarding the reviewer’s questions:
> 1. The “number of tokens changed” refers to the number of tokens that differs from the initial sequence after convergence/termination of the self-correction step. While this roughly correlates with the number of inference steps, it is not a one-to-one correspondence since self-correction may, at times, oscillate between two or more similar states (e.g. multiple options that are equally correct/incorrect). We therefore deem the number of changed tokens after convergence to be a more meaningful metric compared to simply the number of self-correction steps.
> 2. Indeed, due to the continuous-time nature of the GIDD diffusion process, the number of denoising steps can be set arbitrarily high. Due to our chosen parameterization, where the model predicts the fully noise-free data which is then re-noised up to the appropriate level, uniform noise is actually continually injected at low levels, so tokens continue to change throughout. Future work can explore different approaches that avoid this, which may lead to improvements.
>
> [1] Zheng et al., 2023. https://arxiv.org/pdf/2306.05685
>
> ---
>
> **Table 1**
> ||$p_u=0.0$|||$p_u=0.1$|||$p_u=0.2$||
> |-|-|-|-|-|-|-|-|-|
> |Temperature $\tau$|#tokens changed|Entropy||#tokens changed|Entropy||#tokens changed|Entropy|
> |(no self-correction)|0.0|3.13||0.0|3.05||0.0|3.03|
> |0.01|14.5|3.12||5.78|3.04||7.65|3.01|
> |0.05|46.2|3.06||23.7|2.98||28.2|2.94|
> |0.1|62.6|3.12||44.8|2.94||58.4|2.87|
> |0.5|52.4|3.13||37.8|2.95||60.2|2.87|
> |1.0|41.4|3.12||35.3|2.96||51.1|2.89|
>
> **Table 2**
> |Model|Clarity|Grammaticality|Factuality|Writing style|Creativity|
> |-|-|-|-|-|-|
> |GIDD ($p_u = 0.0$)|2.51|2.96|3.61|2.84|4.48|
> |+ self-correction ($\tau = 0.1$)|1.99 (-20.9%**)|2.39 (-19.3%**)|3.02 (-16.2%**)|2.24 (-21.1%**)|3.60 (-19.5%**)|
> |GIDD ($p_u = 0.1$)|2.51|2.85|3.66|2.78|4.26|
> |+ self-correction ($\tau = 0.1$)|2.69 (+7.2%**)|3.05 (+6.9%**)|3.88 (+6.0%**)|2.98 (+7.1%**)|4.35 (+2.1%*)|
> |GIDD ($p_u = 0.2$)|2.49|2.82|3.70|2.79|4.25|
> |+ self-correction ($\tau = 0.5$)|2.90 (+16.5**)|3.29 (+16.6%**)|4.01 (+8.5%**)|3.16 (+13.4%**)|4.48 (+5.5%**)|
>
> Significance levels: * $>2\sigma$ difference, ** $>5\sigma$ difference.

---

> > ### Comment · Reviewer_wojY · 2025-04-03
> >
> > Thanks for the author's reply!
> >
> > After checking Table 2, I’m keeping my promise: *“If the author provides the aforementioned more detailed evaluation metrics (or at least some of them), I will increase my score.”* Since the other two reviewers gave a score of 3, I’ve decided to raise mine from 2 to 4 because I hope the paper gets accepted.
> >
> > However, I still have some concerns. The entropy values reported by the authors appear to be unusually low. According to Section 6.1 in [1], the normal entropy range is approximately 5.6–5.7. Moreover, if the authors could pre-train or fine-tune a larger model to show the method’s performance on math or code tasks, I would definitely be inclined to highlight this paper.
> >
> > [1] Zheng et al. Masked Diffusion Models are Secretly Time-Agnostic Masked Models and Exploit Inaccurate Categorical Sampling. ICLR 2025.
> >
> > -----------------------
> > **After Reply Rebuttal Comment by Authors**
> >
> > I would like to clarify that **entropy is computed directly on the generated tokens and is unrelated to the choice of the reference model.** The authors claim that the poor entropy results are due to their choice of reference model, which is confusing. I believe the authors must have made a mistake here, so I have adjusted my score to a 3.

---

> > > ### Author Response · Authors · 2025-04-04
> > >
> > > We thank the reviewer for their rebuttal comment and for the positive response to the additional experiments.
> > >
> > > Regarding the lower entropy compared to [1], this stems from the fact that we use Gemma-2-9B as our reference model, implying that absolute numbers are not comparable between ours and [1]. The reason we choose Gemma-2-9B over GPT2-large is that we find the latter to be too small of a model to constitute a reasonable approximation of the true distribution of natural language. For the same reason, absolute gen. PPL numbers are also not comparable between ours and prior work using GPT2-large as a reference model. This is in addition to the numerical instabilities of Gumbel sampling, highlighted in [1] and also Appendix H, making our numbers not comparable to much of the existing MDM literature.
> > >
> > > ---
> > >
> > > **EDIT (April 8):** Thank you to the reviewer for pointing out our oversight regarding how **entropy is computed directly on the generated tokens**. While we believe that the generative entropy reported in our initial rebuttal (Table 1) is still a useful metric, it is indeed not how entropy is commonly computed in the literature. Following [1] (App. H.1), we have redone the entropy calculation via sequence-level unigram modeling, the result of which is reported in Table 1 below. We still observe an expected but moderate drop in entropy after self-correction. Hybrid models ($p_u>0$) actually have higher entropy scores already before self-correction, indicating greater sample diversity.
> > >
> > > We hope that the reviewer will still see this update and reconsider their final score accordingly.
> > >
> > > [1] Zheng et al., 2024. https://arxiv.org/abs/2409.02908
> > >
> > > **Table 1**
> > > ||$p_u=0.0$||$p_u=0.1$||$p_u=0.2$||
> > > |-|-|-|-|-|-|-|
> > > |Temperature $\tau$|#tokens changed|Entropy|#tokens changed|Entropy|#tokens changed|Entropy|
> > > |(no self-correction)|0.0|4.95|0.0|5.09|0.0|5.08|
> > > |0.01|14.5|4.67|5.78|5.04|7.65|5.03|
> > > |0.05|46.2|4.54|23.7|4.97|28.2|4.94|
> > > |0.1|62.6|4.66|44.8|4.99|58.4|4.94|
> > > |0.5|52.4|4.83|37.8|5.01|60.2|4.98|
> > > |1.0|41.4|5.06|35.3|5.10|51.1|5.09|

---

### Official Review · Reviewer_LBxr · 2025-03-13

**Overall Recommendation:** 3

**Summary:**

This paper generalizes discrete diffusion models with masked or uniform transition kernels to a larger design space. Specifically, the authors introduce a Generalized Interpolating Discrete Diffusion process (GIDD), which transfers data not to the [mask] state but to an arbitrary predefined distribution. They further prove the existence of such transition kernels and provide ELBO for GIDD. This paper especially focuses on mixing the masked and the uniform noising schemes and demonstrates the empirical advantage of GIDD against existing non-autoregressive methods of language modeling.

**Claims And Evidence:**

Theoretical and empirical contributions are claimed in extending the discrete diffusion framework, which is well supported by evidence.

**Essential References Not Discussed:**

Related works are well discussed in this paper.

**Experimental Designs Or Analyses:**

The reviewer checked the experimental design and analyses. The implementation of GIDD with $p_u=0.0$ resembles MDM, which is consistent with the theoretical claims. The additional uniform noise slightly worsens the PPL of GIDD, as shown in Table 3 and Table 5, which seems to weaken the claim that combining masking and uniform noise improves sample quality. The authors also analyze the proposed self-correction sampling method, showing better self-correction ability of GIDD with additional uniform noise.

**Methods And Evaluation Criteria:**

The proposed method, GIDD, extends discrete diffusion models to generate categorical data. It is evaluated on language generation (by perplexity) and accuracy over downstream tasks, which is an effective evaluation criterion. The effectiveness of the proposed hybrid noising process is evaluated by performing self-correction sampling and computing the generative perplexity under Gemma 2 9B.

**Other Comments Or Suggestions:**

See above.

**Other Strengths And Weaknesses:**

#### Strengths

- This paper provides a richer design space for discrete diffusion models that would be intriguing to explore for future works. This generalized framework is already a contribution by itself.
- The proposed GIDD process is rigorously formulated and well articulated.

#### Weaknesses

- It seems that the additional uniform noise on top of the masked diffusion process does not improve the overall performance of the discrete diffusion model, which undermines the meaningfulness of the hybrid noising process.
- The advantage of GIDD with $p_u>0$ is discussed in Section 5.4, but the importance of self-correction is not well explained. For example, does this self-correcting scheme improve downstream benchmark accuracy for GIDD with $p_u>0$?

**Questions For Authors:**

Some of my points listed in weaknesses may be inaccurate, and I would be grateful for any clarification. Specifically, what is the importance of self-correcting and why is it not demonstrated in Table 2~4?

**Relation To Broader Scientific Literature:**

This paper may be of interest to a broader audience in other scientific domains, such as protein generation with discrete diffusion models.

**Theoretical Claims:**

The paper characterizes the Markov chain for the GIDD process and shows the ELBO of GIDD. The reviewer checked the proofs of Proposition 3.3, Lemma 3.6, and Theorem 3.7 and believes that the proofs are correct.

---

> ### Author Rebuttal · Authors · 2025-03-31
>
> We thank the reviewer for their insightful and detailed review and for acknowledging the theoretical contributions of GIDD.
> In the following, we would like to shed some additional light on the empirical evaluation of the proposed model and, hopefully, address the reviewer’s concerns.
>
> Broadly speaking, our evaluation (and evaluation of language models in general) consists of two parts. The first part is likelihood-based evaluation, which tests the model’s ability to _recognize_ well-formed, high-likelihood sentences, and includes PPL on the test set as well as downstream multiple-choice benchmark performance. The second part is to test the model’s ability to _generate_ well-formed, high-likelihood sentences and for unconditional generation, this is most commonly measured with generative-PPL. This ability is equally, if not more, important because it is how language models are usually used in practice. And while the two abilities may correlate in general, sampling from the model can sometimes introduce challenges not present when simply evaluating likelihood. For example, teacher-forcing is used both when training autoregressive models and when evaluating their likelihood, but not when generating samples, which makes them prone to self-induced mistakes [1]. The same holds true for mask-only diffusion models. The point of our experiments is to show that while, indeed, hybrid diffusion models find it more challenging to evaluate the likelihood of given samples, they have self-correction capabilities and are more robust/less prone to self-induced errors at sampling time, thus generating higher-quality samples overall and especially so for low inference-compute budgets (see Sec. 5.4, L435 ff.; Fig 6, App. D). To further bolster this point, we provide an additional LLM-based evaluation in our response to Reviewer wojY, where we quantify the improvements in sample quality during the self-correction step in terms of clarity, grammaticality, factuality, writing style, and creativity. Perhaps the distinction between _recognition_ and _generation_ should be highlighted and discussed more prominently in the paper, which we will be happy to do in an updated version.
>
> Regarding the second weakness: The reason we cannot naively apply self-correction to the multiple-choice benchmarks is because these benchmarks rely on likelihood-based answer selection. Specifically, for a given question (or prompt) with a set of possible answers (or continuations), the one with the highest likelihood under the model is selected. Since this process does not involve sampling from the model, it is a priori not possible to utilize the self-correction capabilities. As a result, benchmark scores of $p_u > 0$ models are hampered in correlation with their worse likelihood (see Tab. 5, App. B). Future work may aim to close the gap by using different criteria for selecting the correct answers (e.g. self-accuracy instead of likelihood), but unfortunately, this was beyond the scope of this project. Alternatively, future work may extend the proposed models to conditional generation, which will allow evaluation on generative benchmarks like GSM8k or HumanEval, where the generative strength and self-correction capabilities of hybrid models may bring substantial improvements over mask-only diffusion models.
>
> [1] Bachmann & Nagarajan, 2024. https://arxiv.org/abs/2403.06963

---

### Decision · Program_Chairs · 2025-05-01

**Decision:**

Accept (poster)

**Comment:**

Masked diffusion models are inherently constrained by the inability to revise tokens once they are generated. To overcome this limitation, the paper presents General Interpolating Discrete Diffusion (GIDD), which adopts a more flexible noise design to enable iterative refinement of tokens. The effectiveness of GIDD is demonstrated through a series of experiments.

The paper received three reviews. In response to the concerns raised in the initial evaluations, the authors submitted a rebuttal that included additional experimental results and clarifications, effectively addressing the key issues. Following the rebuttal and subsequent discussions, all three reviewers reached a consensus, leading to a decision to weakly accept the paper.

This agreement is attributed to the paper's innovation, profound theoretical justification, and sufficient empirical validation that it offers for this particular study. The AC agrees with the reviewers and recommends accepting the paper. To further improve the paper quality, AC suggests the authors to revise their paper by taking into account all the suggestions provided by the reviewers, including integrating extra experimental results conducted during the rebuttal phase.